# Why Tree-Style Branching Matters for Thought Advantage Estimation in GRPO

**Hongcheng Wang** [* 1 2]   **Yinuo Huang** [* 3 2]   **Sukai Wang** [4]   **Guanghui Ren** [+ 4]   **Hao Dong** [✉ 1 2]

## Abstract

Group Relative Policy Optimization (GRPO) trains Chain-of-Thought reasoning with verifiable rewards, but estimating thought-level advantages without value functions often suffers from high variance. Although tree-style branching is used in practice to reduce variance, it lacks a theoretical explanation of why it works and whether it is important or potentially necessary. We study thought-level advantage estimation in GRPO from a variance perspective under a minimal tree-style setting where multiple continuations are sampled for each thought. Using the multivariate delta method, we reveal a sampling-dimension asymmetry. Increasing sampled thoughts ($K$) leaves a strictly positive estimation-variance floor, whereas increasing continuations per thought ($M$) drives the leading-order estimation variance to zero at rate $1/M$. This implies that, within the fixed-temperature GRPO-style estimator without value models studied here, accurate thought-level advantage estimation cannot be achieved by scaling thought sampling alone, making continuation-level branching a principled and potentially necessary mechanism rather than a heuristic. Experiments further provide empirical evidence for its effectiveness and potential necessity, demonstrating improved optimization stability, training efficiency, and final performance not only in math but also across vision domains and under different model architectures and sizes.

## 1. Introduction

The DeepSeek-R1 model demonstrates that reinforcement learning (RL)—particularly Group Relative Policy Optimization (GRPO) (Shao et al., 2024)—is effective for training Chain-of-Thought (CoT) reasoning. GRPO prompts the LLM to generate a reasoning trace before producing the final answer and then reinforces this process via verifiable rewards. Subsequent methods such as DAPO (Yu et al., 2025), Dr.GRPO (Liu et al., 2025), and GPG (Chu et al., 2025) refine GRPO's loss function from different perspectives, achieving more stable training and stronger mathematical reasoning. Beyond text-based tasks, the GRPO paradigm has also expanded to multi-modal domains (Chen et al., 2025; Shen et al., 2025; Huang et al., 2025b; Feng et al., 2025; Song et al., 2025; Kim et al., 2025), where task-specific reward designs are adopted to improve performance under diverse objectives. Collectively, these results demonstrate that CoT combined with verifiable RL rewards substantially enhances reasoning across modalities.

Although GRPO has demonstrated strong empirical performance across a wide range of domains, it still suffers from instability in thought-level advantage estimation. In standard GRPO-style algorithms, each thought is typically evaluated using a single sampled answer, which contributes to high-variance advantage estimates and unstable gradient updates during training. A growing body of recent work (Kazemnejad et al., 2024; Li et al., 2025; Yang et al., 2025; Guo et al., 2025) empirically adopts tree-style branching to improve the stability of GRPO-style training. However, most of these tree-style methods primarily demonstrate effectiveness through empirical results only on mathematical tasks, rely on complex heuristic segmentation or branching rules, and lack a principled theoretical explanation of *why* branching improves learning. More importantly, they do not address whether such branching is merely a helpful heuristic or necessary for accurate thought advantage estimation when using GRPO-style, group-normalized advantage estimators without value models.

In this work, we view a tree-style branching point as splitting a generation into a *shared context* and one or more *continuations*: the context is the prefix whose advantage is estimated by aggregating continuation rewards. We instantiate this view at the thought–answer boundary by delimiting each thought with an XML tag (*e.g.*, `</think>`) and sampling multiple answers conditioned on that thought, yielding **GRPO-MA (GRPO with Multi-Answer)**.

---

[*]Equal contribution [+]Project Lead [1]CFCS, School of Computer Science, Peking University, Beijing, China [2]PKU–Agibot Joint Lab, Beijing, China [3]University of Electronic Science and Technology of China, Chengdu, China [4]Agibot, Beijing, China. Correspondence to: Hao Dong <hao.dong@pku.edu.cn>.

*Proceedings of the $43^{rd}$ International Conference on Machine Learning*, Seoul, South Korea. PMLR 306, 2026. Copyright 2026 by the author(s).

Using the multivariate delta method, we analyze shared-context advantage estimation under GRPO-style group normalization and **reveal a fundamental *asymmetry* between sampling dimensions** (within a first-order delta-method approximation under an independence assumption). Increasing the number of shared contexts ($K$) alone cannot eliminate context-conditional reward noise: the estimation variance converges to **a strictly positive constant**. In contrast, increasing the number of continuations per shared context ($M$) **drives the leading-order shared-context advantage estimation variance to zero at rate** $1/M$. For GRPO-MA, this means that sampling multiple answers per thought is a principled and *potentially necessary* mechanism for accurate thought-level advantage estimation *under GRPO-style estimators without value models*.

Beyond theory, our experiments with GRPO-MA provide direct empirical evidence for the *effectiveness* of answer-level branching, complementing the variance analysis that motivates its *potential necessity*. Comparing T4A1[1] and T4A4, answer-level branching consistently improves performance across diverse domains. Comparing T4A4 with a thought-scaling baseline T16A1 reveals a better efficiency–performance trade-off: T4A4 runs at substantially lower computational cost while *matching or slightly outperforming* T16A1, showing that answer-level branching is a principled design choice, not merely an efficiency optimization. T4A4 also exhibits significantly fewer gradient spikes than T16A1 (Gradient Spike Score, GSS), consistent with our analysis that lower-variance estimation stabilizes optimization. These benefits are robust across tasks, model sizes, and architectures. Our contributions are summarized as follows:

- **An explanation for why branching helps.** Our delta-method analysis of shared-context advantage estimation reveals a fundamental *asymmetry* (under first-order and independence assumptions): continuation-level branching drives the leading-order shared-context advantage estimation variance to zero at rate $1/M$, whereas context-level branching alone leaves a strictly positive first-order estimation-variance floor—explaining branching as a principled, potentially necessary mechanism rather than a heuristic.

- **GRPO-MA as a lightweight instantiation to test the explanation.** To examine the explanation concretely, we instantiate the framework as GRPO-MA, which branches at explicit delimiters such as `</think>` or `</analysis>` and samples multiple answers per thought with minimal changes to standard GRPO.

---

[1] $TKAM$ denotes a sampling configuration with $K$ thoughts and $M$ post-thought continuations per thought; in result tables, TN and AN report these two counts. $M=1$ corresponds to standard GRPO-style sampling, while $M>1$ corresponds to GRPO-MA. We use this notation throughout the experiments.

- **Empirical evidence supporting the analysis across tasks and model configurations.** Experiments across math, code, vision, and embodied manipulation provide direct empirical evidence for the analysis, showing that continuation-level branching yields consistent gains over standard GRPO and a better efficiency–performance trade-off than context (thought) scaling.

**Conflict of Interest Disclosure.** Several authors are affiliated with Agibot or the PKU–Agibot Joint Lab (see author affiliations), and our object detection experiments use the publicly released AgiBot World dataset (contributors, 2024), developed by Agibot. The core claims are also evaluated on tasks that do not use AgiBot World. No other financial conflicts of interest exist with respect to the methods or benchmarks evaluated.

## 2. Related Work

### 2.1. GRPO Stabilization via Objective and Gradients

A line of work improves the stability and efficiency of GRPO-style training by revisiting the objective, advantage computation, and gradient estimation. DAPO (Yu et al., 2025) combines several practical techniques, including exploration-aware clipping (Clip-Higher), dynamic filtering of uninformative samples (Dynamic Sampling), and a token-level policy-gradient loss to better weight long reasoning traces. Dr. GRPO (Liu et al., 2025) mitigates response-length and question-difficulty biases by removing specific normalization terms in the loss and advantage calculations. GPG (Chu et al., 2025) simplifies the GRPO objective and introduces gradient rescaling to reduce the impact of "zero-gradient" samples. Related methods further target stability from complementary angles: GSPO (Zheng et al., 2025) realigns importance sampling at the sequence level, GMPO (Zhao et al., 2025) uses geometric-mean aggregation to reduce sensitivity to outliers, and GTPO (Simoni et al., 2025) analyzes trajectories to resolve gradient conflicts and prevent policy collapse.

### 2.2. Tree Search for LLM Reinforcement Learning

Recent research has shifted from independent chain-based rollouts to tree-structured sampling to enhance exploration efficiency and credit assignment. VinePPO (Kazemnejad et al., 2024) performs Monte Carlo value estimation through multiple suffix rollouts. SPO (Guo et al., 2025) generalizes the concept of segmentation, enabling flexible-granularity advantage estimation. TreeRL (Hou et al., 2025) introduces an entropy-guided search strategy that expands the search at high-uncertainty tokens. TreePO (Li et al., 2025) estimates advantages based on the collective outcomes of grouped sub-trees to improve robustness. TreeRPO (Yang et al., 2025) defines tree nodes at the level of reasoning steps

and derives step-wise supervision by propagating outcome rewards across sibling branches.

A key distinction among prior tree-search methods lies in where branching is introduced along the generation process. In this work, we adopt semantic XML tags (*e.g.*, `</think>`) as branching points, which provide a simple and implementation-friendly instantiation that supports the variance analysis in Sec. 3.

# 3. Method and Analysis

We recall GRPO, define shared-context advantage estimation, derive its GRPO-style variance, and then instantiate it as GRPO-MA at the thought–answer boundary.

## 3.1. GRPO Preliminaries

GRPO (Shao et al., 2024) is a PPO-style algorithm (Schulman et al., 2017) that computes advantages by normalizing rewards from $K$ sampled responses. For a prompt $p$, GRPO generates responses $\{o_i\}_{i=1}^K$ with rewards $\{R_i\}$ and computes $A(o_i) = \frac{R_i - \text{Mean}(\{R_k\})}{\text{Std}(\{R_k\})}$. The complete GRPO objective can be written as:

$$
\begin{aligned}
&\mathcal{J}_{\text{GRPO}}(\theta) \\
&= \mathop{\mathbb{E}}_{\substack{(p,a)\sim\mathcal{D}\\ o\sim\pi_{\theta_{\text{old}}}}} \left[ \frac{1}{K} \sum_{i=1}^K \frac{1}{T_i^o} \sum_{t=1}^{T_i^o} \ell_{\text{clip}}(r_t, A(o_i)) \right] \\
&\quad - \beta\, D_{\text{KL}}(\pi_\theta \| \pi_{\text{ref}}).
\end{aligned}
\tag{1}
$$

where $T_i^o$ denotes the length of the $i$-th output trajectory, $r_t = \frac{\pi_\theta(o_{i,t}|p,o_{i,<t})}{\pi_{\theta_{\text{old}}}(o_{i,t}|p,o_{i,<t})}$ is the per-token likelihood ratio, and $\ell_{\text{clip}}(r, A) \triangleq \min(rA, \text{clip}(r, 1\pm\varepsilon)A)$ is the clipped surrogate term.

Eq. (1) shows that the advantage controls both the direction and scale of token-probability updates; lower-variance advantages therefore stabilize policy updates.

## 3.2. Shared-Context Advantage Estimation

Eq. (1) makes the advantage estimator the channel through which sampling noise enters the GRPO update; the GRPO primitive we reuse below is its group $z$-score map from sampled returns to normalized advantages. Shared-context advantage estimation applies this same normalization map after replacing each complete-response reward with a context value $V(s)$ estimated by averaging $M$ continuations conditioned on the same shared context.

**Definition 3.1** (Shared-Context Advantage Estimation). Let a generation be split into a *shared context s* and a *continuation a*. *Shared-context advantage estimation* estimates the advantage of $s$ by sampling $M$ continuations $a_1, \ldots, a_M \sim$

$\pi_\theta(\cdot \mid s)$ and forming $V(s) = \frac{1}{M} \sum_{j=1}^M R(s, a_j)$. Given a group of $K$ shared contexts $s_1, \ldots, s_K$, each sampled with its own $M$ continuations and value $V(s_i)$, the per-context advantages $\{A(s_i)\}_{i=1}^K$ are obtained by normalizing the value vector $\{V(s_i)\}_{i=1}^K$. In the GRPO-style estimator analyzed below, this normalization is the group $z$-score $A(s_i) = (V(s_i) - \bar{V})/S_V$, where $\bar{V}$ and $S_V$ are the empirical mean and standard deviation across the $K$ shared contexts. GRPO-MA instantiates this definition by choosing the breakpoint at `</think>` or `</analysis>`; we return to this operational thought–answer form in Sec. 3.4.

## 3.3. Variance of Shared-Context Advantage

### 3.3.1. SETUP AND ASSUMPTIONS

Fix a prompt $p$ and sample the $K$ shared contexts from Def. 3.1 as $s_i \overset{\text{i.i.d.}}{\sim} \pi_\theta(\cdot \mid p)$. For each context, sample continuations $a_{i,j} \sim \pi_\theta(\cdot \mid p, s_i)$ and score them by $R_{i,j} := r(p, s_i, a_{i,j})$.

Let $\mu_{R_i} := \mathbb{E}[R_{i,j} \mid p, s_i]$ and $\sigma_{R_i}^2 := \text{Var}(R_{i,j} \mid p, s_i)$, where $\{R_{i,j}\}_{j=1}^M$ are i.i.d. conditional on $(p, s_i)$. For the GRPO-MA instantiation in Sec. 3.4, this noise is non-trivial: fixing the thought prefix and sampling $M{=}4$ post-thought completions, the mean per-thought reward variance on Math reaches 38.5% of the Bernoulli upper bound (App. G.6).

The empirical value $V(s_i)$ defined in Def. 3.1 then satisfies $\mathbb{E}[V(s_i) \mid p, s_i] = \mu_{R_i}$ and $\text{Var}(V(s_i) \mid p, s_i) = \sigma_{R_i}^2/M$. Throughout, $\widehat{\text{Var}}_{\text{delta}}[A(s_i) \mid s_{1:K}]$ denotes the first-order multivariate delta-method estimation variance over continuation sampling, conditional on sampled contexts $s_{1:K}$ (equivalently, on $\{\mu_{R_k}, \sigma_{R_k}^2\}$); all shared-context advantage variance expressions in this analysis refer to this object. For the large-$K$ analysis we assume mild regularity: positive population shared-context value variance, positive focal conditional reward variance, uniformly bounded conditional reward variances ($\sup_k \sigma_{R_k}^2 < \infty$), and a non-outlier focal context ($|\tilde{\mu}_i| = O(1)$).

**Independence assumption.** Because the shared contexts are drawn i.i.d. ($s_i \sim \pi_\theta(\cdot \mid p)$) and, conditional on each context, the continuations are i.i.d., the estimators $\{V(s_i)\}_{i=1}^K$ are independent by the sampling design, so the covariance matrix is diagonal. Mild residual correlations may still arise from implementation-level effects of sharing the same prompt. Appendix C.5 reports a sanity check for such residual implementation-level correlations in the GRPO-MA thought-level instantiation.

### 3.3.2. APPROXIMATE VARIANCE ESTIMATION

**Intuition and proof sketch.** For fixed $K$ and $\sigma_{\mu_R}^2 > 0$, conditional CLT gives first-order covariance $\text{diag}(\sigma_{R_k}^2/M)$

for $\{V(s_k)\}_{k=1}^K$. Let $g_i(V)$ denote the GRPO-style group standardization map from $\{V(s_k)\}$ to $A(s_i)$. Linearizing around $\{\mu_{R_k}\}$ gives $A(s_i) \approx g_i(\mu) + \sum_k \frac{\partial g_i}{\partial V(s_k)}\big|_\mu \big(V(s_k) - \mu_{R_k}\big)$, where

$$\frac{\partial g_i}{\partial V(s_k)}\bigg|_\mu = \frac{1}{\sigma_{\mu_R}}\left(\delta_{ik} - \frac{1}{K} - \frac{\tilde{\mu}_i \tilde{\mu}_k}{K-1}\right).$$

Taking the variance of this linearized expression and substituting $\mathrm{Var}(V(s_k) \mid p, s_k) = \sigma_{R_k}^2/M$ yields the leading $O(1/M)$ delta-method term (Oehlert, 1992) (full algebra in App. C):

$$\boxed{\begin{aligned}&\widehat{\mathrm{Var}}_{\mathrm{delta}}[A(s_i) \mid s_{1:K}] \\ &= \frac{1}{M\,\sigma_{\mu_R}^2}\sum_{k=1}^K\left(\delta_{ik} - \tfrac{1}{K} - \tfrac{\tilde{\mu}_i \tilde{\mu}_k}{K-1}\right)^2 \sigma_{R_k}^2\end{aligned}} \quad (2)$$

where $\delta_{ik}$ is the Kronecker delta, $\mu_{\bar{R}} = \frac{1}{K}\sum_{k=1}^K \mu_{R_k}$, $\sigma_{\mu_R}^2 = \frac{1}{K-1}\sum_{k=1}^K(\mu_{R_k} - \mu_{\bar{R}})^2$, and $\tilde{\mu}_i = \frac{\mu_{R_i} - \mu_{\bar{R}}}{\sigma_{\mu_R}}$. By definition $\widehat{\mathrm{Var}}_{\mathrm{delta}}$ is the leading $O(1/M)$ term of the multivariate delta-method expansion; the omitted higher-order terms of the true conditional variance vanish as $M \to \infty$ with $K$ fixed. Operationally, the group baseline is analogous to a control-variate baseline: with $M{=}1$, a single reward mixes shared-context quality with idiosyncratic continuation noise; with $M{>}1$, averaging conditionally-independent continuations preserves prefix-dependent value while suppressing continuation noise, producing the $1/M$ variance-reduction term.

Because Eq. (2) is derived for generic shared contexts rather than a specific thought–answer split, it applies to any breakpoint defining $K$ contexts and $M$ continuations under the stated assumptions. This is an algebraic invariance of the estimator, not a claim that all breakpoints are equally effective; empirical quality still depends on conditional reward variance, granularity, and cost. App. C.6 maps representative tree-style methods into the same notation.

### 3.3.3. CONTROLLABLE AND UNCONTROLLABLE SOURCES OF VARIANCE.

The analysis is conducted at a *fixed sampling temperature*: temperature influences $\sigma_{R_i}^2$, but lowering it also changes the target $V(s) = \mathbb{E}_{a \sim p_\theta(\cdot|s)}[R(s,a)]$ (App. G.7). Thus our controllable levers are $K$ and $M$; mechanisms that alter $\{\mu_{R_k}, \sigma_{R_k}^2\}$ are complementary to continuation-level branching rather than substitutes.

### 3.3.4. ANALYSIS OF THE VARIANCE STRUCTURE

Combining the above results, we compare, *within the first-order delta-method approximation of Eq.* (2), increasing

shared contexts $K$ versus continuations $M$. By the law of large numbers,

$$\sigma_{\mu_R}^2 = \frac{1}{K-1}\sum_{k=1}^K(\mu_{R_k} - \mu_{\bar{R}})^2,$$

converges almost surely to the population variance

$$\sigma_\pi^2 = \mathrm{Var}_{s \sim \pi_\theta(\cdot|p)}\big[E[V(s) \mid p, s]\big],$$

which captures true shared-context value variability under policy $\pi_\theta$ and prompt $p$. For a fixed focal index $i$ with $|\tilde{\mu}_i| = O(1)$, the $k = i$ term in Eq. (2) converges to a non-zero constant, while the uniformly bounded $k \neq i$ terms vanish at rate $O(1/K)$. Thus, for fixed $M$,

$$\widehat{\mathrm{Var}}_{\mathrm{delta}}[A(s_i) \mid s_{1:K}] \xrightarrow[K \to \infty]{\text{a.s.}} \frac{\sigma_{R_i}^2}{M\,\sigma_\pi^2} > 0,$$

which is strictly positive under the regularity conditions ($\sigma_\pi^2 > 0$ and $\sigma_{R_i}^2 > 0$). Thus, increasing $K$ alone cannot remove the context-conditional reward noise term when $M$ is fixed.

In contrast, for any fixed $K$ and sampled contexts, increasing $M$ directly suppresses the diagonal covariance $\mathrm{Var}(V(s_i) \mid p, s_i) = \sigma_{R_i}^2/M$ that enters Eq. (2); for the leading delta-method term,

$$\widehat{\mathrm{Var}}_{\mathrm{delta}}[A(s_i) \mid s_{1:K}] = O\left(\frac{1}{M}\right) \xrightarrow[M \to \infty]{} 0,$$

since at fixed $K$ and fixed sampled contexts the leading delta-method term is exactly proportional to $1/M$. Within this first-order approximation, these results reveal a fundamental asymmetry: increasing $K$ alone gives no predictable finite-$K$ reduction and only approaches a positive variance floor, whereas increasing $M$ drives the leading-order conditional estimation variance to zero at the controllable rate $1/M$ (exactly monotone in $M$ at fixed $K$, within this first-order approximation). For GRPO-MA, this asymmetry motivates multiple post-thought continuations per thought as a potentially necessary ingredient for stabilizing thought-level advantage estimation in the fixed-temperature GRPO-style setting analyzed here.

### 3.4. GRPO-MA: Thought–Answer Instantiation

GRPO-MA applies shared-context advantage estimation at the semantic thought–answer boundary: $s_i$ is the thought prefix ending at `</think>` or `</analysis>`, and $a_{i,j}$ is the answer continuation; for Math, the continuation spans the `<process>` and `<answer>` segments after `</analysis>` (Sec. 4). This gives a concrete, lightweight instantiation of the variance principle above.

As shown in Fig. 1, GRPO-MA samples $K$ thoughts $\{th_i\}$ as in GRPO and then samples $M$ answers $\{ans_{i,j}\}_{j=1}^M$ per

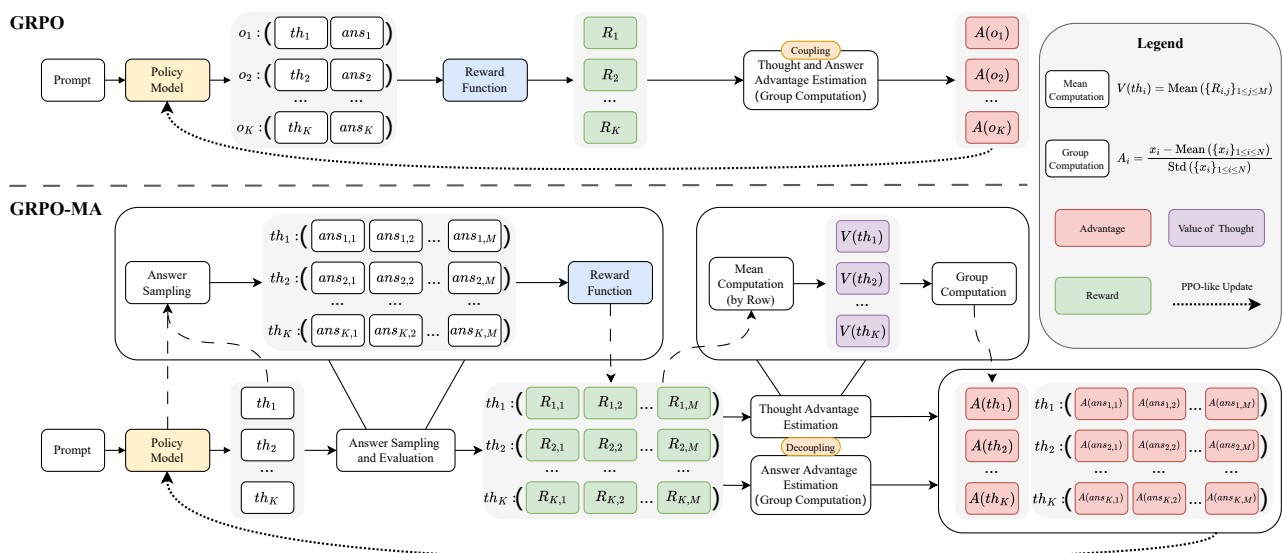

*Figure 1.* **The operational flow of advantage estimation in GRPO and GRPO-MA**. In the baseline GRPO framework (top), the advantage is computed from a single thought–answer pair, coupling thought and answer advantages to one reward signal. GRPO-MA (bottom) samples multiple answers for each thought.

thought. Given rewards $\{R_{i,j}\}$, it computes $V(th_i) = \frac{1}{M}\sum_j R_{i,j}$, then normalizes $\{V(th_i)\}$ for thought advantages and all rewards $\{R_{i,j}\}$ for answer advantages: $A(th_i) = \frac{V(th_i) - \text{Mean}(\{V(th_k)\})}{\text{Std}(\{V(th_k)\})}$ and $A(ans_{i,j}) = \frac{R_{i,j} - \text{Mean}(\{R_{k,l}\})}{\text{Std}(\{R_{k,l}\})}$.

The GRPO-MA objective then combines the two levels of advantages under the joint sampling distribution $\mathcal{S}_{\text{MA}}$, where $(p, a^\star) \sim \mathcal{D}$, $th_{1:K} \overset{\text{i.i.d.}}{\sim} \pi_{\theta_{\text{old}}}(\cdot \mid p)$, and, for each $i$, $ans_{i,1:M} \overset{\text{i.i.d.}}{\sim} \pi_{\theta_{\text{old}}}(\cdot \mid p, th_i)$:

$$
\mathcal{J}_{\text{GRPO-MA}}(\theta)
$$
$$
= \underset{\mathcal{S}_{\text{MA}}}{\mathbb{E}} \left[ \frac{1}{K}\sum_i \frac{1}{T_i^{\text{th}}} \sum_{t=1}^{T_i^{\text{th}}} \ell_{\text{clip}}\big(r_t, A(th_i)\big) \right.
$$
$$
\left. + \frac{1}{KM}\sum_{i,j} \frac{1}{T_{i,j}^{\text{ans}}} \sum_{t=1}^{T_{i,j}^{\text{ans}}} \ell_{\text{clip}}\big(r_t, A(ans_{i,j})\big) \right]
$$
$$
- \beta\, D_{\text{KL}}(\pi_\theta \| \pi_{\text{ref}}). \tag{3}
$$

where $a^\star$ is the dataset reference/verifier signal, and $T_i^{\text{th}}$ and $T_{i,j}^{\text{ans}}$ are thought and post-thought continuation lengths. The thought- and continuation-level terms enter with equal unit weight; the remaining notation follows GRPO.

## 4. Experiments

We evaluate GRPO-MA on Math (Yu et al., 2025), Code (PrimeIntellect, 2024; White et al., 2024), several distinct vision tasks (Object Detection (contributors, 2024), Affordance Prediction (Myers et al., 2015; Luo et al., 2022), Trajectory Prediction (Ji et al., 2025), Demand Prediction (Wang et al., 2024), OCR-based VQA (Biten et al., 2019; Tito et al., 2021)) and a Simulator-based Visual Manipulation task (Li et al., 2024). Our experiments use Qwen2.5-VL-3B-Instruct (Bai et al., 2025) as the base model, with all training conducted on four H100 80G GPUs using LoRA (Hu et al., 2022) for parameter-efficient fine-tuning. For each task, we conduct a group of experiments separately.

**More details (tasks, datasets, hyperparameters, training settings) are in Appendices D and E.**

### 4.1. Text and Vision Task

#### 4.1.1. TASK SETTING AND METRIC

Table 1 summarizes evaluation metrics for text and vision tasks. These tasks collectively cover a wide range of modalities and reasoning types, enabling a comprehensive evaluation of GRPO-MA.

For **Math**, we adopt a structured output format using `<analysis>`, `<process>`, and `<answer>` tags; GRPO-MA applies multi-sampling to the post-`</analysis>` continuation spanning both `<process>` and `<answer>`. For all other tasks, multi-sampling is applied only to `<answer>`.

We track the **Gradient Spike Score (GSS)** (Huang et al., 2025a) to measure gradient stability, defined as $\text{GSS}(g_i) =$

*Table 1.* Evaluation metrics for each task.

| Task | Evaluation Metric |
|------|-------------------|
| **Math** | Pass@k (Chen et al., 2021) |
| **Code** | Pass@k |
| **Object Detection** | Accuracy: proportion of predictions with IoU > threshold. |
| **Affordance Prediction** | Accuracy: proportion of correctly matched points. |
| **Trajectory Prediction** | DFD (Eiter & Mannila, 1994), HD (Huttenlocher et al., 2002), RMSE, EndPoint Dist. |
| **Demand Prediction** | Accuracy: proportion of correct points. |
| **OCR-based VQA** | ANLS (Biten et al., 2019) |

$\frac{|g_i|}{\frac{1}{T+1}\sum_{j=0}^{T}|g_j|}$, where $g_j$ represents the gradient at the j-th time step. We report the number of spikes above 10 (GSS@10), where smaller is better.

### 4.1.2. BASELINES

We adopt models from the Qwen2.5-VL-Instruct series (3B, 7B, and 72B) (Bai et al., 2025) as baselines to evaluate the performance of general-purpose models on our tasks. In addition, we train Qwen2.5-VL-3B-Instruct with real labels using supervised fine-tuning (SFT) to compare against GRPO, denoted as **SFT** in the results. Finally, we compare our proposed GRPO-MA with GRPO under different numbers of responses to demonstrate the superiority of GRPO-MA in terms of training efficiency and performance.

### 4.1.3. MAIN RESULTS ON TEXT AND VISION TASK

The experimental results are presented in Tab. 2 (Math and Code), Tab. 3 (Object Detection, Affordance Prediction, and Demand Prediction), and Tab. 4 (OCR-based VQA and Trajectory Prediction). Among trained Qwen2.5-VL-3B variants, GRPO-MA consistently improves over the GRPO and SFT baselines across tasks, demonstrating strong versatility across diverse domains. At the Pass@1 level, the gap to T4A1 is largest on the easier benchmarks (GSM8K +8.04, HumanEval +2.56) and consistent but smaller on the harder AIME and LiveBench, where absolute scores remain low for all baselines; this benchmark-difficulty pattern is discussed in App. G.8. Case studies are included in Appendix F.

**Gradient Stability.** While our theoretical analysis focuses on the variance of thought-level advantage estimation, its implications for optimization stability follow from the structure of the GRPO-MA objective in Eq. 3, where advantage estimates play a central role in shaping gradient updates. By reducing advantage variance through answer-level branching, GRPO-MA indirectly suppresses extreme gradient magni-

tudes and leads to more stable optimization. Consistent with this implication, T4A4 achieves the lowest GSS@10 in most experimental settings, indicating a substantially reduced frequency of gradient spikes during training. Additional grad_norm and GSS curves are provided in Appendix G.5.

**Performance and Efficiency.** We analyze efficiency from two complementary perspectives: (1) From a per-step sampling cost perspective, normalizing by the cost of T4A1 reveals a scaling pattern across tasks. Extending sampling along the answer dimension incurs only a modest increase in per-step wall-clock cost (typically on the order of 1.1–1.3× relative to T4A1), whereas increasing the number of independently sampled thoughts leads to a substantially larger computational overhead (approximately 1.8–2.4×). Importantly, despite operating at a significantly lower per-step cost, answer-level branching (T4A4) largely matches the thought-scaling baseline T16A1 on final performance while clearly improving over T4A1 at lower cost than thought scaling. (2) Beyond per-step cost, we further evaluate efficiency from an end-to-end wall-clock perspective. As shown by the wall-clock time curves in Appendix G.4, GRPO-MA reaches peak performance earlier than baselines and attains the highest final performance using less training time. This indicates that the improved optimization stability brought by reduced variance in thought-level advantage estimation not only avoids excessive per-step overhead in T16A1, but also accelerates effective progress during training in practice.

These results indicate that the efficiency gains of GRPO-MA do not arise from trading off performance for train speed. Instead, they stem from a more effective allocation of additional samples across sampling levels: by concentrating extra sampling on the answer dimension, GRPO-MA yields more stable advantage estimates, allowing improved optimization behavior to translate into both faster training and stronger performance. Crucially, in our experiments the same efficiency–performance trade-off is not matched by expanding thought samples alone (consistent with the first-order variance analysis), making answer-level branching a principled — and, under fixed-temperature GRPO-style estimators without value models, potentially necessary — design choice for reliable variance reduction and efficient optimization under GRPO-style training.

### 4.2. Simulator-based Manipulation Task

#### 4.2.1. TASK SETTING

We largely follow the experimental protocol of ManipLLM (Li et al., 2024), a simulator-based framework for visual manipulation. To make the task more challenging, we introduce two changes. First, to increase observational diversity, the camera viewpoint is randomly sampled in each trial. Second, we adopt a stricter success criterion: an attempt is immediately counted as a failure if the predicted

*Table 2.* Combined results for Math and Code generation benchmarks on Qwen2.5-VL-3B-Instruct. TN/AN: number of thoughts and answers per thought; S/S: seconds per training step. For AIME and LiveBench, cells report Pass@1/Pass@10/Pass@32 with $N{=}100$ samples per problem. GSM8K and HumanEval report Pass@1 with $N{=}10$. Bold marks the best result among the GRPO variants for each metric; dashes indicate metrics not evaluated for that baseline. Full grad-norm / GSS curves are in App. G.5.

| Model | TN | AN | | | Math | | | | | Code | |
| | | | S/S | GSS@10 | AIME P@1 / P@10 / P@32 | GSM8K P@1 | S/S | GSS@10 | LiveBench P@1 / P@10 / P@32 | HumanEval P@1 |
| --- | --- | --- | --- | --- | --- | --- | --- | --- | --- | --- |
| Qwen2.5-VL-3B-Ins | – | – | – | – | – / 9.27 / 16.25 | – | – | – | – / 9.80 / 11.67 | – |
| Qwen2.5-VL-7B-Ins | – | – | – | – | – / 9.97 / 18.39 | – | – | – | – / 10.72 / 11.31 | – |
| Qwen2.5-VL-72B-Ins | – | – | – | – | – / 33.07 / 41.39 | – | – | – | – / 20.39 / 22.37 | – |
| SFT | – | – | – | – | – / 11.07 / 18.11 | – | – | – | – / 8.72 / 10.59 | – |
| GRPO | 4 | 1 | 111.24 | **5** | 2.07 / 11.78 / 20.32 | 62.54 | 76.21 | **6** | 6.59 / 11.56 / 13.70 | 56.10 |
| GRPO | 8 | 1 | 140.05 | 13 | 2.20 / 11.16 / 21.30 | 64.67 | 104.83 | 24 | 6.53 / 11.44 / 13.39 | 56.83 |
| GRPO | 16 | 1 | 225.43 | 15 | 2.93 / 12.89 / 21.72 | 65.13 | 186.91 | 25 | 6.30 / **11.92** / 14.12 | 57.80 |
| GRPO-MA | 4 | 4 | 132.87 | **5** | **3.40 / 14.70 / 27.60** | **70.58** | 93.45 | 10 | **6.73** / 11.69 / **14.70** | **58.66** |

*Table 3.* Combined Results for Object Detection, Affordance, and Demand Prediction. TN: The number of thoughts; AN: The number of answers per thought; S/S: Second/Step during training; UMD: UMD Part Affordance Dataset; AGD20K: AGD20K Dataset; Bold indicates the best performance among models of the same size; near-ties below one percentage point are not bolded.

| Model | TN | AN | | | Object Detection | | | | | | Affordance Prediction | | | Demand Prediction | |
| | | | S/S | GSS@10 | IoU@0.5 | IoU@0.6 | IoU@0.7 | IoU@0.8 | S/S | GSS@10 | UMD | AGD20K | S/S | GSS | Accuracy |
| --- | --- | --- | --- | --- | --- | --- | --- | --- | --- | --- | --- | --- | --- | --- | --- |
| Qwen2.5-VL-3B | | | | | 60.87 | 50.54 | 39.67 | 21.32 | | | 38.98 | 52.73 | | | 11.41 |
| Qwen2.5-VL-7B | | | | | 70.11 | 60.23 | 48.02 | 25.89 | | | 34.65 | 43.29 | | | 19.95 |
| Qwen2.5-VL-72B | | | | | 72.57 | 60.66 | 47.19 | 24.48 | | | 59.13 | 60.59 | | | 26.27 |
| SFT | | | | | 64.63 | 54.73 | 42.43 | 22.96 | | | 66.35 | 53.18 | | | 36.97 |
| GRPO | 4 | 1 | 13.86 | 5 | 65.11 | 56.16 | 43.88 | 23.02 | 14.34 | 11 | 78.91 | 55.60 | 13.74 | **5** | 38.13 |
| GRPO | 8 | 1 | 18.86 | 30 | 67.13 | 57.52 | 42.62 | 22.82 | 18.92 | 14 | 88.14 | 57.90 | 18.20 | 12 | 40.81 |
| GRPO | 16 | 1 | 26.99 | 16 | 69.03 | 60.29 | 45.16 | 24.04 | 24.27 | 22 | 89.32 | 57.24 | 25.42 | 37 | 42.47 |
| GRPO-MA | 4 | 4 | 15.77 | **1** | **69.71** | **61.32** | **46.77** | **25.64** | 15.86 | **5** | **89.96** | **58.40** | 14.33 | 6 | 42.63 |

contact point is not on the target object surface. As in ManipLLM, when the model predicts a grasp point in the image, we execute a rule-based grasping routine. The suction cup approaches the predicted point along the local surface normal, and we further adjust the subsequent trajectory based on the object category. We evaluate performance by the success rate of predicted points that result in successful manipulation.

### 4.2.2. BASELINES

We adapt some of the baselines used above and add several additional baselines. **ManipLLM-7B:** ManipLLM collects a large number of successful samples in the simulator and constructs multiple task-specific question–answer pairs, using an SFT training approach. We fine-tune its weights under our new setting. **CoT-SFT:** We collect successful samples of GRPO-MA-T4A4 (including the chain of thoughts and answers), then fine-tune Qwen2.5-VL-3B using SFT. **GRPO-NoThink:** We employ GRPO to train the Qwen2.5-VL-3B, but we do not require the model to generate a thought process; instead, it directly produces the answers.

### 4.2.3. MAIN RESULTS ON MANIPULATION TASK

Manipulation tasks with sparse rewards constitute an extreme high-variance regime for GRPO-style training[2], where instability in thought-level advantage estimation is particularly pronounced. The experimental results are presented in Tab. 5. A direct comparison shows that T4A4 significantly outperforms T4A1, indicating that answer-level branching is especially effective in this setting by stabilizing training and facilitating the discovery of successful responses under sparse supervision. This result further supports our variance-based analysis, suggesting that tree-style branching becomes particularly important for reliable credit assignment in high-variance manipulation tasks.

Furthermore, our experiments shed light on the complementary role of the Chain of Thought (CoT) in this context. We observe that the GRPO-NoThink variant, which removes the CoT while sampling the same number of answers as GRPO-MA-T4A4, suffers a substantial performance degradation. Together with the strong performance of the CoT-

---

[2]For simplicity, we assume a binary reward $R \in \{0, 1\}$ that follows a Bernoulli distribution with success probability $p$. When $p$ is very small, reward observations are extremely noisy, making single-sample estimates unreliable and leading to high variance in thought-level advantage estimation.

*Table 4.* Combined Results for OCR-based VQA and Trajectory Prediction. TN: The number of thoughts; AN: The number of answers per thought; S/S: Second/Step during training. Bold indicates the best performance among models of the same size; near-ties below one percentage point are not bolded.

| Model | TN | AN | OCR-based VQA | | | | | Trajectory Prediction | | | | |
|---|---|---|---|---|---|---|---|---|---|---|---|---|
| | | | S/S | GSS@10 | Infographics | St VQA | Doc VQA | S/S | GSS@10 | DFD | HD | RMSE | EndPoint |
| Qwen2.5-VL-3B | | | | | 73.10 | 67.63 | 91.33 | | | 571.60 | 537.63 | 404.40 | 429.93 |
| Qwen2.5-VL-7B | | | | | 78.94 | 74.03 | 93.51 | | | 496.44 | 451.19 | 340.33 | 354.03 |
| Qwen2.5-VL-72B | | | | | 79.72 | 74.27 | 93.26 | | | 386.83 | 352.18 | 263.61 | 300.77 |
| SFT | | | | | 74.77 | 69.68 | 92.94 | | | 277.68 | 261.86 | 196.55 | 228.62 |
| GRPO | 4 | 1 | 14.79 | 42 | 73.70 | 68.94 | 93.15 | 29.17 | 14 | 187.99 | 172.58 | 140.80 | 142.74 |
| GRPO | 8 | 1 | 19.62 | 88 | 76.33 | 71.56 | 93.98 | 34.51 | 18 | 172.41 | 157.09 | 130.09 | 137.29 |
| GRPO | 16 | 1 | 26.79 | 68 | 76.65 | 72.25 | 94.20 | 66.55 | 21 | 165.16 | 149.59 | 122.95 | 130.56 |
| GRPO-MA | 4 | 4 | 17.17 | **17** | 76.69 | 72.48 | 94.22 | 35.41 | **10** | **151.10** | **138.29** | **111.59** | **120.60** |

*Table 5.* Manipulating Point Prediction. TN: The number of thoughts; AN: The number of answers per thought; S/S: Second/Step during training.

| Model | TN | AN | Success Rate (%) | |
|---|---|---|---|---|
| | | | Seen | Unseen |
| Qwen2.5-VL-3B | | | 4.73% | 1.30% |
| ManipLLM-7B | | | 22.80% | 7.63% |
| SFT | | | 9.17% | 4.28% |
| CoT-SFT | | | 28.18% | 11.79% |
| GRPO | 4 | 1 | 10.75% | 3.94% |
| GRPO-NoThink | 0 | 16 | 10.60% | 2.40% |
| GRPO-MA | 4 | 4 | **31.40%** | **16.00%** |

SFT model, this indicates that a high-quality CoT is a necessary prerequisite for generating semantically meaningful thoughts, while answer-level branching is essential for reducing the variance of their downstream credit assignment. This division of roles clarifies why both components are critical for effective learning in complex manipulation tasks with sparse rewards.

### 4.3. More Baseline Comparison

*Table 6.* More Baseline Comparison on Trajectory Prediction .

| Model | DFD | HD | RMSE | EP |
|---|---|---|---|---|
| GRPO | 187.99 | 172.58 | 140.80 | 142.74 |
| GRPO-CARE | 188.23 | 170.82 | 139.83 | 144.85 |
| DAPO | 184.08 | 167.75 | 136.18 | 146.99 |
| Dr.GRPO | 180.56 | 165.95 | 135.86 | 140.05 |
| TreeRPO-128 | 165.27 | 149.77 | 123.42 | 131.45 |
| TreeRPO-64 | 157.03 | 144.52 | 113.99 | 125.08 |
| **GRPO-MA** | **151.10** | **138.29** | **111.59** | **120.60** |

To provide a broader comparison, we evaluate four recent GRPO-based variants—GRPO-CARE, DAPO, Dr.GRPO, and TreeRPO—on the trajectory prediction task. As shown in Table 6, GRPO-MA achieves the best performance across all metrics. GRPO-CARE, DAPO, and Dr.GRPO improve GRPO via loss or normalization refinements but do not introduce explicit branching, so thought-level advantages are still estimated from single outcomes and remain high-variance under sparse rewards. TreeRPO relies on token-length–based branching, where the suffix (*e.g.*, -64 or -128) denotes the maximum segment length used for splitting responses. As a result, its behavior is sensitive to response length: TreeRPO-128 degenerates to standard GRPO when responses are short, resulting in poorer performance compared to TreeRPO-64. Although TreeRPO-64 mitigates this issue and improves performance relative to TreeRPO-128, it still underperforms GRPO-MA, which we conjecture is due to higher-variance estimation of values (or advantages) for thoughts in sub-branches induced by branching at intermediate segments. In contrast, our XML-tag–based branching operates on semantic boundaries, enabling stable branching and reducing variance in thought-level advantage estimation across variable-length outputs.

### 4.4. The Effectiveness on Larger Model Size

To study how GRPO-MA behaves as model capacity increases, we run a scaling experiment on Qwen2.5-VL-7B-Instruct. The 7B scale is a common "large" checkpoint for tree-search and multi-sampling reasoning methods (Guo et al., 2025; Kazemnejad et al., 2024; Li et al., 2025), balancing model capacity against rollout cost. We compare GRPO-MA with a matched GRPO baseline under identical settings on trajectory prediction (Table 7). GRPO-MA still outperforms GRPO, indicating that multi-answer sampling continues to benefit larger models.

*Table 7.* Qwen2.5-VL-7B-Instruct on Trajectory Prediction.

| Model | DFD | HD | RMSE | EP |
|---|---|---|---|---|
| 7B-T8A1 | 167.32 | 152.34 | 129.50 | 134.60 |
| 3B-MA-T4A4 | 151.10 | 138.29 | 111.59 | 120.60 |
| **7B-MA-T8A4** | **134.67** | **121.89** | **103.64** | **103.27** |

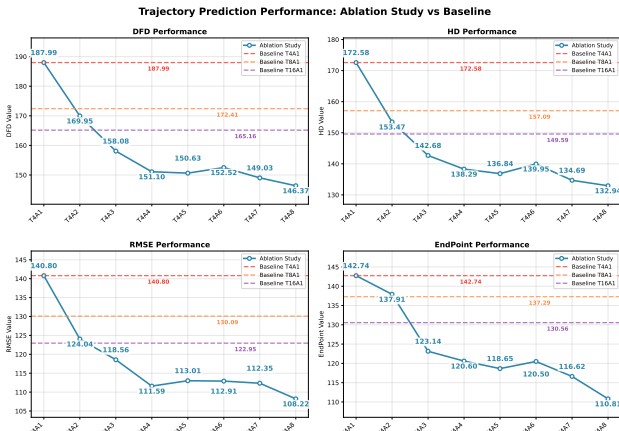

*Figure 2.* **Ablation Study on Trajectory Prediction** While maintaining the number of thoughts $K = 4$, we gradually increase the number of responses $M$ per thought from 1 to 8 (*i.e.*, the number of responses is 4, 8, 12...32).

### 4.5. The Effectiveness on Different Model Architecture

We further evaluate GRPO-MA on the pure-text Qwen3-4B-Instruct for math and code, keeping the training configuration unchanged. Table 8 shows GRPO-MA-T4A4 matches GRPO-T16A1 on math Pass@1 (18.07 vs 17.90) at only $\sim 55\%$ of its per-step cost.

*Table 8.* Qwen3-4B-Instruct on Math and Code. S/S: seconds per training step (math). The code-side run for T16A1 was omitted under our compute budget.

| Model | S/S | Pass@1(N=10) on Math | Pass@1(N=10) on Code |
|---|---|---|---|
| GRPO-T4A1 | 115.28 | 17.53 | 4.22 |
| GRPO-T16A1 | 245.83 | 17.90 | **6.25** |
| **GRPO-MA-T4A4** | 135.42 | 18.07 | **6.25** |

### 4.6. Ablation Study

We ablate the number of answers per thought $M$ on the trajectory prediction task (Fig. 2). As $M$ increases, most evaluation metrics decline, with a progressively smaller rate of decline.

Surprisingly, T4A3 (4 thoughts, 3 answers each, 12 total) outperforms T16A1 (16 single-answer thoughts) across all metrics. A plausible reason is that thought quality matters more than reward-signal quantity: scoring each thought by $V(th_i) = \frac{1}{M} \sum_{j=1}^{M} R_{i,j}$ with $M = 3$ yields a more stable estimate than T16A1's single-sample evaluation ($M = 1$), which is more sensitive to reward noise.

## 5. Conclusion

We studied tree-style branching in GRPO-style training as *shared-context advantage estimation*, where a shared prefix is evaluated through multiple continuations. The delta-method analysis reveals an asymmetry: increasing continuations per context suppresses context-conditional reward noise, whereas increasing independently sampled contexts alone leaves a first-order variance floor. This explains why reusing a prefix with multiple continuations can be more effective than simply sampling more complete trajectories.

GRPO-MA instantiates this idea at the thought–answer boundary using delimiters such as `</think>` and `</analysis>`, staying close to standard GRPO with no value model or step-level tree construction. Experiments across math, code, vision, and embodied manipulation show improved stability, efficiency, and final performance at lower sampling cost. The analysis holds under the stated assumptions and the validation focuses on this breakpoint; future work can study adaptive breakpoints, variance-aware continuation allocation, and combinations with value modeling or pruning.

## Acknowledgements

This work was supported by National Natural Science Foundation of China (62376006) and National Natural Science Foundation of China (62136001). We thank the reviewers and area chair for their constructive feedback on the camera-ready version.

## Impact Statement

This paper presents work whose goal is to advance the field of Large Language Models. There are many potential societal consequences of our work, none of which we feel must be specifically highlighted here.

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

## A. Appendix Outline

Below is the table of contents for the appendix.

## B. More Related Work: Applications of GRPO in Multimodal Domains

VLM-R1 (Shen et al., 2025) applies a general GRPO pipeline to Vision-Language Models, enabling smaller models to achieve competitive performance on complex visual reasoning tasks. Vision-R1 (Huang et al., 2025b) generates high-quality multimodal Chain-of-Thought data and uses Progressive Thinking Suppression Training (PTST) to prevent the model from creating overly long reasoning paths. Video-R1 (Feng et al., 2025) introduces Temporal-GRPO (T-GRPO), a novel reward scheme that encourages the model to leverage temporal information in video sequences. ManipLVM-R1 (Song et al., 2025) employs GRPO for robotic manipulation with new affordance-aware and trajectory matching reward functions to improve the localization of interactive parts and the physical plausibility of actions. Robot-R1 (Kim et al., 2025) reframes robot learning as a multiple-choice question answering task, using GRPO to optimize the reasoning for embodied manipulation.

# C. Full Analysis of Shared-Context Advantage Variance

This section provides a full derivation of the approximate variance for the shared-context advantage $A(s_i)$, as presented in the main paper. We first review the multivariate Delta Method, establish the asymptotic normality of the shared-context value estimators via the Central Limit Theorem (CLT), and finally present the detailed application and gradient calculation. In GRPO-MA, the shared context $s_i$ is the thought prefix ending at `</think>` or `</analysis>`, and the continuation $a_{i,j}$ is the sampled answer (the `<process>`–`<answer>` span after `</analysis>` for the Math task); the thought-advantage formula used by GRPO-MA is therefore a direct instantiation of this result.

## C.1. The Multivariate Delta Method

The Delta Method is a fundamental result in statistics used to approximate the moments of a function of one or more random variables. The multivariate version is central to our analysis.

**General Formulation.** Let $\overrightarrow{V} = (V_1, V_2, \ldots, V_K)$ be a $K$-dimensional random vector of estimators with a true mean vector $\overrightarrow{\mu} = (\mu_1, \mu_2, \ldots, \mu_K)$. Let $M$ denote the sample size used to compute each estimator $V_k$. To emphasize that these estimators are functions of the sample size, we denote the vector as $\overrightarrow{V}_M$. The Delta Method provides the asymptotic distribution of $f(\overrightarrow{V}_M)$ as $M \to \infty$. Specifically, if $\overrightarrow{V}_M$ satisfies the condition for the Central Limit Theorem such that:

$$\sqrt{M}(\overrightarrow{V}_M - \overrightarrow{\mu}) \xrightarrow{d} N(0, \Sigma_{\text{asymptotic}}) \tag{4}$$

where $\xrightarrow{d}$ denotes convergence in distribution, then the transformed variable $f(\overrightarrow{V}_M)$ also converges in distribution:

$$\sqrt{M}(f(\overrightarrow{V}_M) - f(\overrightarrow{\mu})) \xrightarrow{d} N(0, \nabla f(\overrightarrow{\mu})^T \Sigma_{\text{asymptotic}} \nabla f(\overrightarrow{\mu})) \tag{5}$$

From this formal result, we derive the practical formula for approximating the variance of $f(\overrightarrow{V}_M)$ for a large but finite sample size $M$. The term $\sqrt{M}$ acts as a scaling factor that ensures the limiting distribution has a finite, non-zero variance. The variance of the estimator itself is given by:

$$\text{Var}(f(\overrightarrow{V}_M)) \approx \nabla f(\overrightarrow{\mu})^T \, \text{Var}(\overrightarrow{V}_M) \nabla f(\overrightarrow{\mu}) \tag{6}$$

where $\text{Var}(\overrightarrow{V}_M)$ is the actual covariance matrix of the estimator vector, which is related to the asymptotic covariance by $\text{Var}(\overrightarrow{V}_M) \approx \Sigma_{\text{asymptotic}}/M$.

## C.2. Asymptotic Normality of the Estimated Value Vector

Before applying the Delta Method, we must first establish that our core estimator, the vector of estimated shared-context values $\overrightarrow{V(s)}$, satisfies the prerequisite of being asymptotically normal. This justification comes from the Central Limit Theorem (CLT).

For each shared context $s_k$, its estimated value $V(s_k)$ is the sample mean of $M$ i.i.d. random variables, the rewards from continuations $\{a_{k,j}\}_{j=1}^{M}$:

$$V(s_k) = \frac{1}{M} \sum_{j=1}^{M} R_{k,j} \tag{7}$$

The rewards have a finite true mean $\mu_{R_k}$ and a finite true variance $\sigma_{R_k}^2$. According to the CLT, as the sample size $M \to \infty$, the distribution of the standardized sample mean converges to a normal distribution. This is formally stated as:

$$\sqrt{M}(V(s_k) - \mu_{R_k}) \xrightarrow{d} N(0, \sigma_{R_k}^2) \tag{8}$$

We now extend this to the full $K$-dimensional vector of estimators, $\overrightarrow{V(s)} = (V(s_1), \ldots, V(s_K))$. Since we have assumed that the estimated values for different shared contexts are mutually independent, the joint asymptotic distribution of the vector is also normal. The mean of this limiting distribution is a zero vector, and the covariance matrix is diagonal, composed of the individual variances. Therefore, the entire vector of estimators is asymptotically normal:

$$\sqrt{M}(\overrightarrow{V(s)} - \overrightarrow{\mu}) \xrightarrow{d} N(0, \Sigma_{\text{diag}}) \tag{9}$$

where $\overrightarrow{\mu} = (\mu_{R_1}, \ldots, \mu_{R_K})$ is the vector of true means, and $\Sigma_{\text{diag}}$ is the diagonal covariance matrix of the limiting distribution:

$$\Sigma_{\text{diag}} = \begin{pmatrix} \sigma_{R_1}^2 & 0 & \cdots & 0 \\ 0 & \sigma_{R_2}^2 & \cdots & 0 \\ \vdots & \vdots & \ddots & \vdots \\ 0 & 0 & \cdots & \sigma_{R_K}^2 \end{pmatrix} \tag{10}$$

This result formally justifies the application of the Multivariate Delta Method to the shared-context advantage function $A(s_i) = f_i(\overrightarrow{V(s)})$.

### C.3. Application to the Shared-Context Advantage Function

**Verification of Assumptions.** The prerequisites for the Delta Method are satisfied. First, as established above, our estimator vector $\overrightarrow{V(s)}$ is asymptotically normal. Second, the advantage function $A(s_i) = (V(s_i) - \bar{V})/S_V$ is continuously differentiable everywhere except where the denominator $S_V = 0$, where $\bar{V} = \frac{1}{K}\sum_{k=1}^{K} V(s_k)$ and $S_V = \sqrt{\frac{1}{K-1}\sum_{k=1}^{K}(V(s_k) - \bar{V})^2}$. We evaluate the gradient at $\overrightarrow{\mu}$, where the denominator's analogue is $\sigma_{\mu_R}$. The approximation is thus valid assuming $\sigma_{\mu_R} > 0$, *i.e.*, not all shared contexts have the same true value.

**Gradient Calculation.** Let $\overrightarrow{V(s)} = V = (V_1, \ldots, V_K)$ and define

$$f_i(V) = A(s_i) = \frac{N_i(V)}{D(V)} = \frac{V_i - \bar{V}}{S_V}, \qquad \bar{V} = \frac{1}{K}\sum_{j=1}^{K} V_j,$$

$$Q(V) = \frac{1}{K-1}\sum_{j=1}^{K}(V_j - \bar{V})^2, \qquad D(V) = S_V = \sqrt{Q(V)}.$$

We compute $\partial f_i / \partial V_k$ in steps.

First,

$$\frac{\partial \bar{V}}{\partial V_k} = \frac{1}{K}, \qquad \frac{\partial N_i}{\partial V_k} = \delta_{ik} - \frac{1}{K}. \tag{11}$$

For $Q$ we have, using $\partial(V_j - \bar{V})/\partial V_k = \delta_{jk} - \frac{1}{K}$,

$$\frac{\partial Q}{\partial V_k} = \frac{1}{K-1}\sum_{j=1}^{K} 2(V_j - \bar{V})\left(\delta_{jk} - \tfrac{1}{K}\right) \tag{12}$$

$$= \frac{2}{K-1}\left[(V_k - \bar{V}) - \frac{1}{K}\sum_{j=1}^{K}(V_j - \bar{V})\right] = \frac{2}{K-1}(V_k - \bar{V}), \tag{13}$$

because $\sum_j (V_j - \bar{V}) = 0$. Therefore

$$\frac{\partial D}{\partial V_k} = \frac{1}{2D}\frac{\partial Q}{\partial V_k} = \frac{V_k - \bar{V}}{(K-1)D}. \tag{14}$$

Applying the quotient rule yields, for arbitrary $V$,

$$\frac{\partial f_i}{\partial V_k} = \frac{(\delta_{ik} - \frac{1}{K})D - (V_i - \bar{V})\cdot \frac{V_k - \bar{V}}{(K-1)D}}{D^2} = \frac{\delta_{ik} - \frac{1}{K}}{D} - \frac{(V_i - \bar{V})(V_k - \bar{V})}{(K-1)D^3}. \tag{15}$$

Evaluate at $V = \overrightarrow{\mu}$ and denote

$$\sigma_{\mu_R} := D\big|_{V=\mu} = \sqrt{\frac{1}{K-1}\sum_j (\mu_j - \bar{\mu})^2}, \qquad \tilde{\mu}_j := \frac{\mu_j - \bar{\mu}}{\sigma_{\mu_R}}.$$

Then

$$\frac{\partial f_i}{\partial V_k}\bigg|_{V=\mu} = \frac{1}{\sigma_{\mu_R}}\left(\delta_{ik} - \frac{1}{K} - \frac{\tilde{\mu}_i\tilde{\mu}_k}{K-1}\right). \tag{16}$$

Finally, by the first-order multivariate Delta method, with $\mathrm{Var}(\overrightarrow{V}) = \frac{1}{M}\Sigma_{\mathrm{diag}}$ (and $\Sigma_{\mathrm{diag}} = \mathrm{diag}(\sigma_{R_1}^2, \ldots, \sigma_{R_K}^2)$),

$$\widehat{\mathrm{Var}}_{\mathrm{delta}}[A(s_i) \mid s_{1:K}] = \nabla_V f_i(\mu)^\top \mathrm{Var}(\overrightarrow{V}) \nabla_V f_i(\mu) = \frac{1}{M\sigma_{\mu_R}^2}\sum_{k=1}^{K}\left(\delta_{ik} - \frac{1}{K} - \frac{\tilde{\mu}_i\tilde{\mu}_k}{K-1}\right)^2\sigma_{R_k}^2. \tag{17}$$

## C.4. Application to the Continuation/Answer Advantage Function

This subsection records an *analogous* first-order linearization for the continuation/answer advantage. It is stated for completeness and is *not* used in the $K$ versus $M$ asymmetry conclusion of the main text. Unlike the thought-level case we do not re-derive the gradient here; moreover, as $M$ grows the index set has growing dimension $KM$, so the expression below should be read as a formal leading-order analogue rather than a rigorous fixed-dimension delta-method statement.

For a single continuation $a_{i,j}$, which is an answer in the GRPO-MA instantiation, the advantage is defined as

$$A(a_{i,j}) = \frac{R_{i,j} - \overline{R}}{S_R}, \quad \overline{R} = \frac{1}{KM}\sum_{k=1}^{K}\sum_{m=1}^{M}R_{k,m}, \quad S_R = \sqrt{\frac{1}{KM-1}\sum_{k=1}^{K}\sum_{m=1}^{M}(R_{k,m} - \overline{R})^2}. \tag{18}$$

Following the same first-order linearization logic, the variance of $A(a_{i,j})$ can be approximated as

$$\mathrm{Var}\big[A(a_{i,j})\big] \approx \nabla_{\mathbf{R}}g_{i,j}(\boldsymbol{\mu})^\top \mathrm{diag}(\sigma_{R_1}^2, \ldots, \sigma_{R_K}^2, \ldots) \nabla_{\mathbf{R}}g_{i,j}(\boldsymbol{\mu}), \tag{19}$$

where $g_{i,j}(\mathbf{R}) = A(a_{i,j})$ and $\boldsymbol{\mu}$ denotes the vector of reward means.

Evaluating the gradient at $\mathbf{R} = \boldsymbol{\mu}$ and grouping by shared context, we obtain

$$\mathrm{Var}\big[A(a_{i,j})\big] \approx \frac{KM-1}{M(K-1)\sigma_{\mu_R}^2}\sum_{k=1}^{K}\sum_{m=1}^{M}\left(\delta_{(k,m),(i,j)} - \frac{1}{KM} - \frac{\tilde{\mu}_i\tilde{\mu}_k}{M(K-1)}\right)^2\sigma_{R_k}^2, \tag{20}$$

where $\delta_{(k,m),(i,j)}$ is the Kronecker delta, $\tilde{\mu}_k = (\mu_{R_k} - \mu_{\bar{R}})/\sigma_{\mu_R}$ is the expected advantage of shared context $s_k$, $\mu_{\bar{R}} = \frac{1}{K}\sum_{k=1}^{K}\mu_{R_k}$, and $\sigma_{\mu_R}^2 = \frac{1}{K-1}\sum_{k=1}^{K}(\mu_{R_k} - \mu_{\bar{R}})^2$.

## C.5. Diagonality Analysis of Matrices

For the GRPO-MA instantiation, although the diagonal covariance in the derivation follows from the idealized sampling design, we empirically examine residual implementation-level correlations among thought-level shared-context value estimates. We conducted numerical simulations and empirically estimated the covariance structure of $\overrightarrow{V}(s) = (V_1, \ldots, V_K)$, where each $s_i$ is a thought prefix. Specifically, we generated $N$ independent replications of the full $K$-dimensional estimator vector, denoted $V^{(n)}$ for $n = 1, \ldots, N$, and computed the empirical covariance matrix:

$$\widehat{\Sigma} = \frac{1}{N-1}\sum_{n=1}^{N}\big(V^{(n)} - \overline{V}\big)\big(V^{(n)} - \overline{V}\big)^\top, \qquad \overline{V} = \frac{1}{N}\sum_{n=1}^{N}V^{(n)}. \tag{21}$$

We then assessed the degree of diagonal dominance using Row-wise strict diagonal dominance and Frobenius-norm based diagonal energy ratio.

**Row-wise strict diagonal dominance.** For each row $i$, the covariance matrix is said to be strictly diagonally dominant if

$$|\widehat{\Sigma}_{ii}| > \sum_{j\neq i}|\widehat{\Sigma}_{ij}|. \tag{22}$$

We summarize this property by the proportion of rows that satisfy the condition:

$$p_{\text{row\_dom}} \;=\; \frac{1}{K}\sum_{i=1}^{K}\mathbf{1}\Big\{|\widehat{\Sigma}_{ii}| > \sum_{j\neq i}|\widehat{\Sigma}_{ij}|\Big\}, \tag{23}$$

where $\mathbf{1}\{\cdot\}$ denotes the indicator function. A value $p_{\text{row\_dom}} \approx 1$ indicates strong diagonal dominance.

**Frobenius-norm based diagonal energy ratio.** We also consider the proportion of squared Frobenius norm explained by the diagonal entries:

$$\rho_F \;=\; \frac{\sum_{i=1}^{K}\widehat{\Sigma}_{ii}^2}{\sum_{i=1}^{K}\sum_{j=1}^{K}\widehat{\Sigma}_{ij}^2}, \qquad 0 \le \rho_F \le 1. \tag{24}$$

Higher values of $\rho_F$ indicate that the diagonal terms dominate the overall covariance energy.

We select 50 samples from the Trajectory Prediction task and, at the 1500-step checkpoint, compute the covariance matrix of the thought-level shared-context value estimates by performing $N = 10$ independent replications per sample. The empirical results yield $p_{\text{row\_dom}}$=**63.65%** and $\rho_F$ = **70.71%** averaged on 50 samples. The structural independence used by our derivation follows from the sampling design (i.i.d. shared contexts; conditionally i.i.d. continuations); this numerical check only probes *residual implementation-level* correlations. The observed $p_{\text{row\_dom}} = 63.65\%$ and $\rho_F = 70.71\%$ indicate approximate, rather than exact, diagonal dominance, consistent with treating residual correlations as approximation error in the first-order diagonal analysis.

### C.6. Unified Mapping to Tree-Style Methods

The generic shared-context derivation in Eq. (2) applies whenever a method generates $K$ independent shared contexts and, conditional on each context, $M$ i.i.d. continuations that are subsequently aggregated by the same GRPO-style group standardization. Tab. 9 instantiates this mapping for three representative tree-style methods.

*Table 9.* Existing tree-style methods cast as shared-context advantage estimation, under the same independence approximation used in Eq. (2). In each case $K$ counts shared contexts and $M$ counts continuations per context; Eq. (2) predicts the same algebraic variance form, while the branch-specific conditional moments $\{\mu_{R_k}, \sigma_{R_k}^2\}$ differ across methods.

| Method | Shared context $s$ | Continuation $a$ |
|---|---|---|
| GRPO-MA (this work) | full thought up to `</think>` (or `</analysis>` for Math) | answer tokens (process+answer for Math) |
| TreePO (Li et al., 2025) | a sub-tree node (variable prefix) | sub-tree leaves |
| TreeRL (Hou et al., 2025) | token prefix up to an entropy split | continuation to terminal |
| TreeRPO (Yang et al., 2025) | token prefix at a step boundary | next step plus continuation |

# D. Details in Task Settings

### D.1. Math

We conduct our experiments using problems from the DAPO (Yu et al., 2025) training set and evaluate on the AIME2024 test set (Maxwell-Jia, 2024). The Math training set is constructed by randomly sampling 1,000 problems from the DAPO training corpus. The model is trained for a single epoch on these 1,000 training samples. We do not use a validation set; instead, we select the final model parameters saved at the end of training (the last checkpoint) for testing.

At test time, for each test problem from AIME2024 we generate $n = 100$ independent candidate outputs ("generations"). From these 100 generations we compute the *pass@k* metrics for $k \in \{1, 10, 32\}$.

The reward function is designed with two complementary components: a *format reward* and an *accuracy reward*. The model is required to generate outputs in a predefined structured format:

```
<analysis> xxx </analysis>
```

```
<process> xxx </process>
<answer> d </answer>
```

where the answer is represented as a single integer $d$. The format reward assigns a value of 1 if and only if the output strictly follows the required format, and 0 otherwise. The accuracy reward is +1 if the predicted answer is identical to the true answer, and 0 otherwise.

The full prompt template is shown below:

```
{Question} You MUST structure your response using exactly
three sections with XML-style tags in this exact order:
1) <analysis> ... </analysis>
2) <process> ... </process>
3) <answer> ... </answer>

Roles and constraints:
- <analysis>: State relevant concepts, theorems, formulas,
and solution plan. Do NOT perform numeric calculations or
write equations here.
- <process>: Perform ALL detailed computations and
step-by-step derivations based on the analysis. Show
equations and numeric work here.
- <answer>: Output ONLY the final integer (optional sign).
No words, units, punctuation (except the sign), or
explanations.

Hard requirements:
- All three tags must be present and appear in the exact
order <analysis> -> <process> -> <answer>.
- No calculations in <analysis>.
- All computations must be in <process>.
- <answer> must contain a single integer only.
```

**Implementation Note**: In our multi-sample framework, the sampled content encompasses both $< answer >$ and $< process >$ elements.

### D.2. Code

We conduct our experiments using the Python-code portion of the SYNTHETIC-1 dataset (PrimeIntellect, 2024) and evaluate on the LiveBench code test set (White et al., 2024). The Code training set is constructed by randomly sampling 1,000 problems from the SYNTHETIC-1 Python-code corpus. The model is trained for a single epoch on these 1,000 training samples. We do not use a validation set; instead, we select the final model parameters saved at the end of training (the last checkpoint) for testing.

At test time, for each test problem from LiveBench we generate $n = 100$ independent candidate outputs ("generations"). From these 100 generations we compute the *pass@k* metrics for $k \in \{1, 10, 32\}$ as described below.

The reward function is designed with two complementary components: a *format reward* and a *functional (accuracy) reward*. The model is required to generate outputs in a predefined structured format:

```
<think> xxx </think>
<answer> xxx </answer>
```

The format reward assigns a value of 1 if and only if the output strictly follows the required tag structure and the content within `<answer>` can be parsed as a syntactically valid Python program. Otherwise the format reward is 0.

The functional (accuracy) reward is +1 if the program inside `<answer>` executes successfully on the official hidden test

inputs, terminates without runtime error, and produces outputs that exactly match the expected outputs for all test cases. Otherwise the accuracy reward is 0.

The prompt used to condition the model for each problem is exactly:

```
{Question} First output the thinking process in <think> </think> tags and
then output the final code in <answer> </answer> tags.  The answer should
be a complete Python code solution that solves the given problem.  Make
sure your code handles all edge cases and follows the input/output format
specified in the problem.  DO NOT OUTPUT ANY CODE OR SOLUTION IN THE THINK
TAGS.
```

### D.3. Object Detection

We conduct our experiments using the **Agibot World dataset** (contributors, 2024). The data is partitioned into training, validation, and test sets based on specific `task_ids` from Agibot World dataset. Specifically, the training set is constructed from `task_ids` 424, 480, and 507, comprising a total of 3,000 images (randomly sampling). The validation and test sets are derived from `task_id` 582 and 1352, respectively. For all images, the ground-truth bounding boxes and corresponding object labels are annotated through a crowdsourcing process. The object detection model is trained for a single epoch on the 3,000-image training set.

After training, we perform model selection by evaluating checkpoints on the designated validation set. The model checkpoint that achieves the highest average IoU@0.5 (as defined below) on the validation data is selected for the final evaluation. The performance of this selected model is then reported on the test set.

We evaluate the model's performance using a **IoU rate** metric, which measures the proportion of correctly localized objects based on the Intersection over Union (IoU). A detection is considered positive if the IoU between the predicted bounding box ($B_{pred}$) and the ground-truth bounding box ($B_{gt}$) exceeds a given threshold $\tau$.

The IoU rate at a specific threshold $\tau$, denoted as IoU@$\tau$, is formulated as:

$$\text{IoU@}\tau = \frac{\sum_{i=1}^{N} \mathbb{I}(\text{IoU}(B_{pred}^{(i)}, B_{gt}^{(i)}) > \tau)}{N} \tag{25}$$

where $N$ is the total number of samples in the test set, and $\mathbb{I}(\cdot)$ is the indicator function. To provide a comprehensive assessment, we report the performance across four different IoU thresholds: $\tau \in \{0.5, 0.6, 0.7, 0.8\}$.

The reward function is designed with two complementary components: a *format reward* and an *accuracy reward*. The model is required to generate outputs in a predefined structured format:

```
<think> xxx </think>
<answer> [d, d, d, d] </answer>
```

where the bounding box is represented as a list of four integers $[d, d, d, d]$. The format reward assigns a value of 1 if and only if the output strictly follows the required format, and 0 otherwise. The accuracy reward is defined as the IoU between the predicted bounding box and the ground-truth bounding box.

The full prompt template is shown below:

```
{Question} First output the thinking process in <think> </think> tags and then
output the final answer in <answer> </answer> tags.  Output the final answer in
List format.  Only output the bounding box using [x_min, y_min, x_max, y_max] format
in the final answer.  DO NOT OUTPUT ANY ANSWER OR CONCLUSION IN THE THINK TAGS.
```

### D.4. Affordance Prediction

The task is defined as affordance prediction, where the model, given an image and a specified affordance (e.g., grasping, holding), is required to predict a pixel-wise mask indicating the corresponding region.

We primarily use the UMD Part Affordance Dataset (Myers et al., 2015). The official training split of this dataset is used to construct our training and validation sets. Specifically, we use 3,000 images for training and a held-out portion of the original training split for validation. For evaluation, we use the official test split of the UMD dataset. To further assess the model's generalization capabilities, we also use the entire AGD20K dataset (Luo et al., 2022) as an additional, challenging test set.

The affordance prediction model is trained for a single epoch on the 3,000-image training set. After training, we perform model selection by evaluating checkpoints on the designated validation set. The model checkpoint that achieves the highest Success Rate (as defined below) on the validation data is selected for the final evaluation. The performance of this selected model is then reported on the test sets (UMD test and AGD20K).

We evaluate the model's performance using a Success Rate metric. This metric measures the proportion of samples where the predicted point correctly falls within the ground-truth affordance mask. A prediction is considered successful if the pixel value at the predicted 2D coordinate is 1 in the ground-truth binary mask.

The Success Rate is formulated as:

$$\text{Success Rate} = \frac{\sum_{i=1}^{N} \mathbb{I}(M_{\text{gt}}^{(i)}(C_{\text{pred}}^{(i)}) = 1)}{N} \tag{26}$$

where $N$ is the total number of samples in the test set, $C_{\text{pred}}^{(i)}$ is the predicted 2D coordinate $(x, y)$ for the $i$-th sample, and $M_{\text{gt}}^{(i)}$ is the corresponding ground-truth affordance mask. The notation $M_{\text{gt}}^{(i)}(C_{\text{pred}}^{(i)})$ represents the value of the mask at the predicted coordinate. $\mathbb{I}(\cdot)$ is the indicator function, which is 1 if the condition is true and 0 otherwise.

The reward function consists of two complementary components: a *format reward* and an *accuracy reward*.

The model is required to generate outputs in the following structured format:

```
<think> xxx </think>
<answer> [d, d] </answer>
```

where the final answer corresponds to a 2D coordinate $[d, d]$, with $d$ denoting an integer. The format reward assigns a value of 1 if and only if the output strictly adheres to this format; otherwise, it is set to 0. The accuracy reward evaluates the correctness of the prediction by checking whether the predicted 2D point lies within the ground-truth affordance mask (i.e., a region where the mask value equals 1). If the prediction falls inside the valid region, +1 reward is given; otherwise, it is not.

The full prompt template is shown below:

```
{Question} First output the thinking process in <think> </think> tags and then
output the final answer in <answer> </answer> tags.  Only output one affordance
point using [x, y] format.  DO NOT OUTPUT ANY ANSWER OR CONCLUSION IN THE THINK
TAGS.
```

### D.5. Trajectory Prediction

The task is defined as trajectory prediction, where the model, given an image and a manipulation instruction, is required to predict the two-dimensional trajectory of the robotic arm's end-effector in the image's pixel coordinate system. The trajectory is represented as a sequence of coordinates, and the predicted path should follow the ground-truth trajectory to successfully complete the instructed manipulation.

We primarily use the trajectory subset of the BAAI ShareRobot dataset (Ji et al., 2025). The original dataset is partitioned into training, validation, and test sets. Specifically, we use 3,000 images for training, a held-out portion of the training split for validation, and the test split for evaluation. The model is trained for a single epoch on the 3,000-image training set. After training, we perform model selection by evaluating checkpoints on the designated validation set. The checkpoint that achieves the highest reward value (as defined below) on the validation data is selected for the final evaluation. The performance of this selected model is then reported on the held-out test set.

We evaluate the model's performance using multiple geometric similarity metrics, following the design in ManipVLM-R1 (Song et al., 2025). These metrics measure how well the predicted trajectory matches the ground truth from different

perspectives. Specifically, we use Discrete Fréchet Distance (DFD), Hausdorff Distance (HD), Root Mean Square Error (RMSE), and Endpoint Distance as evaluation criteria.

The model is required to generate outputs in the following structured format:

```
<think> xxx </think>
<answer> [[x1, y1], [x2, y2], ..., [xn, yn]] </answer>
```

where the final answer corresponds to a variable-length sequence of 2D coordinates $[x, y]$, with $x$ and $y$ denoting integers.

The reward function consists of two complementary components: a *format reward* and a *accuracy reward*. The format reward assigns a value of $1$ if and only if the output strictly adheres to this format; otherwise, it is set to $0$. To measure how well the predicted trajectory $\hat{T}$ matches the ground-truth trajectory $T^*$, we adopt an accuracy reward following the design in ManipVLM-R1 (Song et al., 2025). Specifically, the reward is defined as

$$
\begin{aligned}
R_{\text{acc}} = \ & \exp\big(-k\, D_{\text{DFD}}(\hat{T}, T^*)\big) + \exp\big(-k\, D_{\text{HD}}(\hat{T}, T^*)\big) \\
& + \exp\big(-k\, D_{\text{RMSE}}(\hat{T}, T^*)\big) + \exp\big(-k\, \|\hat{p}_N - p_M^*\|^2\big),
\end{aligned}
\tag{27}
$$

where $D_{\text{DFD}}$, $D_{\text{HD}}$, and $D_{\text{RMSE}}$ denote the Discrete Fréchet Distance, Hausdorff Distance, and Root Mean Square Error between the predicted trajectory $\hat{T}$ and the ground-truth trajectory $T^*$. The final term enforces endpoint accuracy by penalizing the distance between the predicted endpoint $\hat{p}_N$ and the ground-truth endpoint $p_M^*$.

The model is guided by a carefully designed prompt that specifies both the reasoning and the answer requirements. The full prompt template is shown below:

```
{Question} First output the thinking process in <think> </think> tags and then
output the final answer in <answer> </answer> tags.  Output the final answer in
the following JSON format:  [[x1, y1], [x2, y2], ..., [xn, yn]].  Where each
coordinate pair represents a point in the image's pixel space and the center of
the end effector needs to follow the coordinates to complete the task.  Each hand
trajectory includes unknown number of [x, y] coordinate pairs.DO NOT OUTPUT ANY
ANSWER OR CONCLUSION IN THE THINK TAGS.
```

### D.6. Demand Prediction

The task is defined as demand prediction, where the model, given an image and a human demand instruction (e.g., "I am thirsty"), is required to output a two-dimensional coordinate corresponding to an object in the image that fulfills the demand (e.g., a water bottle or a juice box). A prediction is considered correct if the predicted point lies inside the ground-truth segmentation mask of the demanded object.

We construct the dataset for this task based on MO-DDN (Wang et al., 2024), which requires robots to ground a natural demand instruction to objects in the environment. MO-DDN itself is built upon the HSSD scene dataset (Khanna et al., 2024), together with a custom demand–object dataset. To build our data, we randomly sample a demand instruction and pair it with a scene containing a target object that satisfies the demand. We then crop and store the corresponding image, resulting in instruction–image pairs.

Following the original MO-DDN splits, we collect data separately from the training and testing tasks. Specifically, we use 3,000 instruction–image pairs as the training set and 1,000 pairs as the validation set, both sampled from the training tasks. For evaluation, we construct a test set of 5,000 instruction–image pairs sampled from the testing tasks.

We train the model for a single epoch on the training set and perform model selection based on validation accuracy. The checkpoint achieving the highest validation performance is then used for testing, and we report results on the test set.

We evaluate the model's performance using a *Success Rate* metric, defined as the proportion of samples where the predicted coordinate falls within the ground-truth mask of the demanded object. Formally:

$$
\text{Success Rate} = \frac{\sum_{i=1}^{N} \mathbb{I}(M_{\text{gt}}^{(i)}(C_{\text{pred}}^{(i)}) = 1)}{N},
\tag{28}
$$

where $N$ is the number of samples in the test set, $C_{\text{pred}}^{(i)}$ denotes the predicted 2D coordinate $(x, y)$ for the $i$-th sample, and $M_{\text{gt}}^{(i)}$ is the ground-truth binary mask of the demanded object. The notation $M_{\text{gt}}^{(i)}(C_{\text{pred}}^{(i)})$ indicates the mask value at the predicted location. $\mathbb{I}(\cdot)$ is the indicator function that equals 1 if the condition holds and 0 otherwise.

The reward function for training consists of two complementary components: a *format reward* and an *accuracy reward*. The model must output predictions in the following structured format:

```
<think> xxx </think>
<answer> [d, d] </answer>
```

where the final answer corresponds to a 2D coordinate $[d, d]$, with $d$ denoting an integer. The format reward is assigned 1 if the output strictly follows this structure, and 0 otherwise. The accuracy reward is assigned if and only if the predicted coordinate lies within the ground-truth object mask. These two rewards jointly ensure syntactically valid outputs and semantic correctness.

The model is guided by a prompt template that specifies both the thinking process and the final answer format. The full prompt is given below:

```
You are completing a navigation task where you need to detect objects from the
image that fulfill a user's demand.  The user's demand is {Question}.  First
output the thinking process in <think> </think> tags and then output the final
answer in <answer> </answer> tags.  Only output one point using [x, y] format that
represents the target demanded object.  DO NOT OUTPUT ANY ANSWER OR CONCLUSION IN
THE THINK TAGS.
```

### D.7. OCR-based VQA

The task is defined as OCR-based Visual Question Answering (VQA), where the model, given an image containing textual information and a natural language question, is required to output a short natural language answer. The answer must be grounded in the image content and can involve both text extraction and reasoning over visual elements.

We construct the dataset by combining three OCR-based VQA benchmarks: *Document VQA* (Tito et al., 2021), *Infographics VQA* (Tito et al., 2021), and *Scene Text VQA* (Biten et al., 2019). Document VQA focuses on answering questions asked over document images, which may contain printed, typewritten, and handwritten content (*e.g.*, letters, memos, reports). The answers are typically text spans taken verbatim from the document. Infographics VQA considers questions over infographic images containing charts, diagrams, or other structured visual data, where answers are not always explicitly extracted text but can include inferred information. Scene Text VQA consists of natural scene images with embedded text (*e.g.*, storefronts, street signs). The model must jointly leverage OCR reading and visual understanding to answer the questions.

From each of the three training sets, we randomly select 3,000 samples, resulting in a combined training set of 9,000 samples. Additionally, we construct a validation set of 1,500 samples (also drawn from the training splits), while the official validation sets of each benchmark are used as our test set.

The model is trained for a single epoch on the 9,000-sample mixed training set. Model selection is performed based on validation performance, and the checkpoint achieving the highest validation score is reported on the test sets.

The evaluation metric is the *Average Normalized Levenshtein Similarity* (ANLS), which measures the string-level similarity between the predicted and ground-truth answers. ANLS accounts for OCR errors by softly penalizing recognition mistakes. A threshold $\tau = 0.5$ is applied to determine whether a predicted answer is considered valid. Formally, ANLS is defined as:

$$\text{ANLS} = \frac{1}{N} \sum_{i=1}^{N} \left( \max_{1 \leq j \leq M} s(a_{ij}, o_{q_i}) \right), \tag{29}$$

$$s(a_{ij}, o_{q_i}) = \begin{cases} 1 - NL(a_{ij}, o_{q_i}), & \text{if } NL(a_{ij}, o_{q_i}) < \tau, \\ 0, & \text{if } NL(a_{ij}, o_{q_i}) \geq \tau, \end{cases} \tag{30}$$

where $N$ is the number of questions, $M$ is the number of ground-truth answers per question, $a_{ij}$ is the $j$-th ground-truth answer for the $i$-th question $q_i$, and $o_{q_i}$ is the predicted answer. $NL(\cdot)$ denotes the normalized Levenshtein distance.

The reward function consists of a *format reward* and an *accuracy reward*. The model must output answers in the following structured format:

```
<think> xxx </think>
<answer> xxx </answer>
```

The format reward is $1$ if the output strictly follows this structure, and $0$ otherwise. The accuracy reward corresponds to the ANLS score of the predicted answer for the current question.

The model is guided by the following prompt template:

```
{Question} First output the thinking process in <think> </think> tags and then
output the final answer in <answer> </answer> tags.  The answer should be a
natural language text.  The answer should be found in the image.  DO NOT OUTPUT
ANY ANSWER OR CONCLUSION IN THE THINK TAGS.
```

### D.8. Simulator-based Visual Manipulation

The task is defined as a simulator-based visual manipulation problem where, given a single RGB observation of a manipulation scene, the model must specify a contact point $(x, y)$ on the object at which a sucker should attempt to manipulate. The model's output must be grounded in the visual observation and may require reasoning about object geometry, affordances, and reachable contact locations.

We construct the dataset and evaluation splits based on the PartNet Mobility dataset (Xiang et al., 2020) and the ManipLLM experimental setup (following their setting, we use suction cups as end effectors).. For training, we adopt the same 20 training categories as ManipLLM, consisting of 1,043 object instances. Training scenes are generated following the SAPIEN simulator (Xiang et al., 2020) setup and ManipLLM scene configurations. For testing, we use the open-sourced ManipLLM test set, which contains approximately 1,830 successful test samples spanning both Seen and Unseen objects. To better evaluate model generalization to novel viewpoints, we further construct a camera-perturbed test set by modifying each test sample: the camera orientation vector $[0, 0, 0]$ is replaced by $[x, y, z]$ where each of $x, y, z$ is sampled uniformly from the signed interval $\pm[0.2, 0.6]$. This perturbation preserves other scene properties while intentionally stressing viewpoint robustness. In order to simplify control and isolate contact selection, the sucker approach direction in all experiments is fixed to be the surface normal at the chosen manipulation point $(x, y)$.

The required output must follow a strict format consisting of a reasoning trace and a final contact point, written as:

```
<think> xxxx </think>
<answer> (d,d) </answer>
```

The evaluation metric is Success Rate, following ManipLLM's criterion based on the manipulated object's displacement after the scripted sucker motion. Formally, given $N$ trials,

$$\text{SuccessRate} = \frac{1}{N} \sum_{i=1}^{N} \mathbb{I}\{\text{trial}_i \text{ is successful according to ManipLLM's displacement criterion}\},$$

where $\mathbb{I}\{\cdot\}$ is the indicator function. We report Success Rate on the camera-perturbed test sets, and further provide breakdowns by Seen vs. Unseen objects.

The reward function during GRPO training consists of a format reward and a task reward. The format reward is $1$ if the output strictly follows the required structure and $0$ otherwise. The task reward is $1$ if the manipulation attempt succeeds according to ManipLLM's displacement criterion and $0$ otherwise. The overall reward is defined as

$$R_{\text{total}} = R_{\text{format}} + R_{\text{task}},$$

so that only properly formatted and successful outputs receive credit. This ensures that malformed answers cannot be rewarded even if the manipulation itself succeeds.

All experiments are conducted with Qwen2.5-VL-3B as the base model. We train using GRPO for 4,000 optimization steps, selecting checkpoints based on validation success rate. The validation set is constructed by sampling held-out scenes from the same 20 training categories without overlap with the test split. The prompt used in training is as follows:

*Table 10.* Hyperparameters for GRPO training.

| Hyperparameter Group | Parameter | Value |
|---|---|---|
| *Training Configuration* | | |
| | Model | Qwen2.5-VL-3B-Instruct |
| | Optimizer | AdamW |
| | Learning Rate ($\eta$) | $1 \times 10^{-5}$ |
| | Batch Size | 1 |
| | Gradient Accumulation Steps | 1 |
| | Total Training Epochs | 1 |
| | Max Completion Length | 4096 |
| | Data Seed | 42 |
| | Floating Point Precision | bfloat16 |
| | Gradient Checkpointing | true |
| | Flash Attention 2 | true |
| *PEFT (LoRA) Configuration* | | |
| | LoRA Rank ($r$) | 64 |
| | LoRA Alpha ($\alpha$) | 128 |
| | LoRA Dropout | 0.05 |
| *GRPO-specific Configuration* | | |
| | Beta ($\beta$) | 0.04 |
| | Epsilon High ($\epsilon_H$) | 0.28 |
| | Epsilon Low ($\epsilon_L$) | 0.2 |
| *Model Specific Configuration* | | |
| | Freeze Vision Modules | true |

```
 "system":  "You are an intelligent manipulator.  A conversation between
User and Assistant.  The user asks a question, and the Assistant solves
it.  The assistant first thinks about the reasoning process in the mind
and then provides the user with the answer.  The reasoning process and
answer are enclosed within <think> </think> and <answer> </answer> tags,
respectively, i.e.  <think> reasoning process here </think><answer> answer
here </answer>."

"user":  "Specify the contact point (x, y) of manipulating the object.
The camera resolution is:'width':  336, 'height':  336, Output format:
<think>your thinking process</think> <answer>(x,y)</answer>"
```

# E. Details in Training

## E.1. Training Hyperparameters

We summarize the key hyperparameters used in our GRPO training experiments in Tab. 10. The settings are organized into general, training, and LoRA-related categories for clarity.

## E.2. SFT Details

For all Supervised Fine-Tuning (SFT) baselines, we train for 5 epochs. All other settings are kept consistent with GRPO, including the dataset, model selection criteria, and metric calculation.

### E.3. Generation Configuration

Our model is trained using the Hugging Face `transformers` library (version 4.51.3). During inference, we customize the decoding strategy via the `GenerationConfig` class. Specifically, we set `temperature=1.0` and `do_sample=True` to enable stochastic sampling. We also define `stop_strings=["</think>", "</analysis>"]` only when generating thoughts. The remaining parameters are maintained at their default settings.

## F. Case Study and Visualization

To provide a more intuitive and in-depth analysis of our model's performance, this section presents a series of curated case studies and visualizations. These examples encompass a range of key tasks, including object detection (Fig. 3 and Fig. 4) and trajectory prediction (Fig. 5 and Fig. 6). Our aim is to leverage these concrete scenarios to delve into the model's behavior, decision-making logic, and inherent strengths and limitations.

Specifically, in the simulator-based visual manipulation task, we visualize the distribution of the target operation points over multiple sampling attempts in Fig. 7. Green points indicate successful manipulations, while red points represent failures. This visualization demonstrates the robustness of our model.

## G. More Experimental Analysis

In this section, we further present some experimental results, including inconsistency analysis, the accuracy reward curves during training, and an analysis of the richness of reward signal.

### G.1. Inconsistency Analysis

We quantify the inconsistency between thoughts and answers during training. For a thought $th_i$ with $M$ answers, if $\text{sign}(A(th_i)) \neq \text{sign}(A(ans_{i,j}))$, we mark it as inconsistent. The inconsistency rate is defined as $\text{InconsistencyRate} = \frac{1}{KM} \sum_{i=1}^{K} \sum_{j=1}^{M} \mathbb{I}[A(th_i)A(ans_{i,j}) < 0]$, where $\mathbb{I}[\cdot]$ denotes the indicator function, which equals 1 if the condition inside holds and 0 otherwise.

Under the T4A4 setting, the inconsistency rate is **25.65%** for trajectory prediction and **24.83%** for object detection. Notably, this ratio is also indicative for GRPO baselines (T4A1, T8A1, T16A1), even though they do not explicitly generate multiple answers per thought and thus cannot directly compute it, since they share the same generation hyperparameters (*e.g.*, temperature, top-$k$, and top-$p$ sampling). This observation further supports our claim that inconsistency is common in GRPO's training. Moreover, this inconsistency implicitly undermines model training.

### G.2. Accuracy Reward Curve

We present the accuracy reward curves for five visual tasks in Object Detection (Fig. 8), Affordance Prediction (Fig. 9), Demand Prediction (Fig. 10), OCR-based VQA (Fig. 11) and Trajectory Prediction (Fig. 12). During the curve plotting process, we smooth the curve using a moving average method with a window size of 200. The curves demonstrate that T4A4 (red) exhibits performance comparable to that of T16A1 (blue) in the majority of cases, at times showing a marginal advantage.

### G.3. Richness of Reward Signals

For tasks with binary (0–1) rewards, such as Code, Math, Affordance Prediction, Demand Prediction, and Simulator-based Visual Manipulation, we measure the proportion of samples whose total reward is positive, which we refer to as the *NoZeroRate*. Formally, it is defined as

$$\text{NoZeroRate} = \frac{1}{T} \sum_{t=1}^{T} \mathbf{1}\left\{ \left( \sum_{i=1}^{K} \sum_{j=1}^{M} \text{AccR}_{i,j}^{t} \right) > 0 \right\}, \tag{31}$$

where $\mathbf{1}\{\cdot\}$ denotes the indicator function, which equals 1 if the condition holds and 0 otherwise. Here, $T$ is the total number of time steps, $t$ indexes a specific time step, $K$ is the number of thoughts, $M$ is the number of answers per thought, and $\text{AccR}_{i,j}^{t}$ denotes the accuracy reward associated with the $j$-th answer under the $i$-th thought at time step $t$. Intuitively, a

*Table 11.* NoZeroRate on Different Task. TN: The number of thoughts; AN: The number of answers per thought. **Bold** indicates the best performance and *italics* indicate the second-best performance.

|        | TN | AN | Code | Math | Affordance | Demand | Sim Manip |
|--------|----|----|------|------|-----------|--------|-----------|
| GRPO | 4 | 1 | 26.71% | 19.14% | 85.10% | 46.37% | 38.20% |
| GRPO | 8 | 1 | 36.57% | 27.71% | 94.37% | 62.07% | / |
| GRPO | 16 | 1 | **43.71%** | **43.29%** | **97.17%** | **70.20%** | / |
| GRPO-MA | 4 | 4 | *41.86%* | *34.71%* | *96.70%* | *66.47%* | **85.05%** |

higher NoZeroRate indicates a lower proportion of advantage collapse (*i.e.*, all advantage values being zero), and thus a larger fraction of samples contributing effective gradient information.

The statistical results are reported in Tab. 11. We observe that T4A4 achieves the second-highest NoZeroRate across all tasks, only behind T16A1. This result suggests that T4A4 strikes a favorable balance between the number of thoughts and the number of answers per thought, leading to a high probability that at least one sampled answer yields a non-zero reward. In particular, compared with T4A1, T4A4 exhibits a substantially higher NoZeroRate, indicating that generating multiple answers per thought significantly reduces the likelihood of advantage collapse.

This comparison also addresses a potential concern regarding whether sampling multiple answers conditioned on the same thought truly produces diverse solutions, rather than near-duplicate ones. If answers sampled from a single thought were largely identical, then increasing the number of answers per thought would be unlikely to introduce higher NoZeroRate, and configurations such as T4A4 and T4A1 would be expected to exhibit similar NoZeroRates. However, under our high-temperature sampling regime, we observe a clear gap between the NoZeroRates of T4A4 and T4A1. This gap indicates that additional answers sampled from the same thought substantially increase the probability of observing at least one non-zero reward, thereby leading to a higher NoZeroRate. This provides indirect but quantitative evidence that multi-sampling under a fixed thought yields diverse and effective answers within our framework.

### G.4. Wall-clock Time

We provide wall-clock time to further demonstrate the improvement in training efficiency achieved by the GRPO-MA algorithm. We pick two multimodal tasks and one text reasoning task as examples. The results are shown in Fig. 13, Fig. 14, and Fig. 15. As shown in the figure, GRPO-MA achieves the peak performance of various baselines in a shorter time.

### G.5. Grad-norm and GSS Curve

We provide complete Grad Norm and GSS curves, shown in Fig. 16, Fig. 17 and Fig. 18. Smaller fluctuations in the Grad Norm curve and lower GSS values indicate fewer gradient spikes during training, resulting in more stable training. The grad norm curve and GSS curve corresponding to GRPO-MA both exhibit smaller fluctuations and GSS values.

### G.6. Per-Thought Answer-Reward Variance

To verify that $\sigma_{R_i}^2$ in our theory is non-trivial in practice, we directly measure per-thought reward variance on three representative tasks. For each training sample we fix the thought prefix and sample $M=4$ answers at temperature 1.0, then compute the task-specific reward normalized to $[0, 1]$. We report the mean of the per-sample variance and the mean of the per-sample range ($\max - \min$ across the $M=4$ answers). The theoretical upper bound for a Bernoulli$(0.5)$ reward is $0.25$.

*Table 12.* Direct measurement of per-thought answer-reward variance.

| Task | Reward | Mean Variance | Mean Range (max−min) |
|------|--------|---------------|----------------------|
| Math (binary) | $\{0, 1\}$ | **0.0963** | **0.4660** |
| Object Detection (IoU) | continuous | 0.0577 | 0.2972 |
| Trajectory Prediction | continuous | 0.0463 | 0.2158 |

Even on the most stable task (Trajectory Prediction), the best-vs-worst answer under the same thought spans more than $20\%$ of the full reward scale; on Math the mean variance reaches $38.5\%$ of the Bernoulli upper bound. The same thought prefix

routinely leads to substantially different rewards, directly grounding the $\sigma^2_{R_i}$ term in our variance expression.

### G.7. Temperature Ablation on Trajectory Prediction

To verify that the gains of answer-level branching are not merely a consequence of averaging out high-temperature decoding noise, we lower the answer-sampling temperature for the T4A1 baseline and compare it to GRPO-MA-T4A4 (which keeps temperature $= 1.0$). Lowering the temperature does improve T4A1, but the result still falls short of T4A4. This is consistent with the controllable/uncontrollable variance discussion in the main text: lowering temperature changes the estimation target $V(th) = \mathbb{E}_{a \sim p_\theta(\cdot | th)}[R(th, a)]$ rather than reducing the variance of the existing target, and discards distributional information learned during pre-training.

*Table 13.* Lowering the answer-sampling temperature for T4A1 vs. GRPO-MA-T4A4 (Trajectory Prediction).

| Setting | DFD↓ | HD↓ | RMSE↓ | EndPoint↓ |
|---|---|---|---|---|
| GRPO-T4A1 (temp=1.0) | 187.99 | 172.58 | 140.80 | 142.74 |
| GRPO-T4A1 (temp=0.1) | 178.63 | 165.95 | 123.16 | 132.19 |
| **GRPO-MA-T4A4** | **151.10** | **138.29** | **111.59** | **120.60** |

### G.8. Benchmark-Difficulty Pattern in Pass@1 Gains

In Tab. 2, the Pass@1 gap between GRPO-MA-T4A4 and the T4A1 baseline is markedly larger on the easier benchmarks (GSM8K $+8.04$, HumanEval $+2.56$) than on the harder ones (AIME $+1.33$, LiveBench $+0.14$). Two factors explain this. First, the absolute Pass@1 ceiling reached by the underlying 3B-scale model on AIME and LiveBench is itself very low ($< 7\%$), so the absolute headroom for any RL recipe is compressed. Second, the variance-reduction mechanism predicted by Eq. (2) tightens the estimate of $V(th)$ for each thought, which translates into accuracy improvements only when the reward signal on enough thoughts is positive; on extremely hard benchmarks most thoughts still receive zero reward, weakening the link between estimator quality and Pass@1. Pass@10/Pass@32 numbers in the same table show the gains re-emerge once a sufficient number of trajectories per problem is considered.

### G.9. More Results on Code and Math

To further examine the generality of our variance-reduction mechanism, we additionally evaluate GRPO-MA on two widely used mathematical and coding benchmarks: GSM8K (Cobbe et al., 2021) for math reasoning and HumanEval (Chen et al., 2021) for code generation. These experiments use the same training configuration as our main results, with models trained on the DAPO dataset. For GSM8K, we report the Pass@1 accuracy obtained by running the checkpoint trained for mathematical reasoning. For HumanEval, we report Pass@1 and Pass@5 to assess both direct correctness and sampling-based performance. These findings reinforce our claim that answer-level multi-sampling provides a simple, general, and effective variance-reduction strategy that benefits both mathematical reasoning and code generation tasks. The consolidated results are presented in Table 14.

## H. Usage of LLMs

We employ a Large Language Model (LLM) to refine the manuscript, with a focus on correcting grammatical errors and enhancing overall readability.

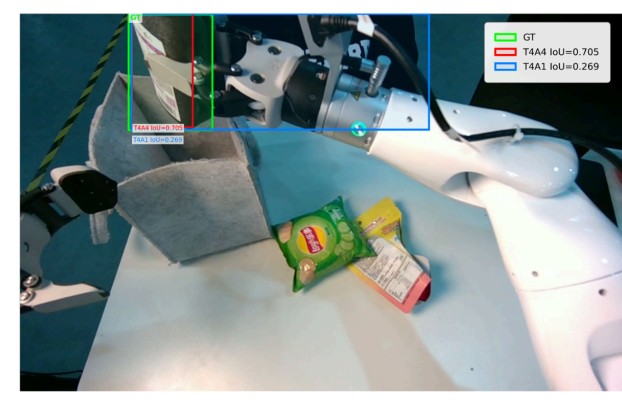

**Prompt:** Please provide the coordinates of the detection box of purple bottled beverage in the image.

**T4A4:** <think>\nThe image shows a robotic arm interacting with objects on a table. The robotic arm is holding a purple bottle, which appears to be a beverage. The bottle is positioned near a green bag and a yellow box. The purple bottle is the only object that matches the description of a purple bottled beverage.\n</think>\n<answer>\n[233, 0, 360, 235]\n</answer>

**T4A1:** <think>\nThe image shows a robotic arm interacting with objects on a table. There is a purple bottled beverage visible in the image, but it is not the main focus. The main objects of interest are the green and yellow packages, which appear to be snacks or food items. The purple bottled beverage is not clearly identifiable in the bounding box coordinates provided.\n</think>\n<answer>\n[232, 0, 853, 240]\n</answer>

**Discussion:** Overall, both T4A4 and T4A1 descriptions are accurate. However, T4A4's description includes more location information about the target object—the purple bottled beverage—while T4A1, though recognizing the target object, mistakenly identified the centrally located object as the main objects.

*Figure 3.* A case study comparing the baseline GRPO with our proposed GRPO-MA on a referring expression grounding task. The prompt is to locate the "purple bottled beverage". The baseline model, GRPO (T4A1), recognizes the target's existence but its reasoning is distracted by other salient objects (the snacks), leading to a failure in grounding. In contrast, our GRPO-MA (T4A4) correctly reasons about the scene's context, focuses on the target object held by the robotic arm, and successfully provides the precise bounding box. This demonstrates the superior robustness of GRPO-MA in complex scene understanding and reasoning.

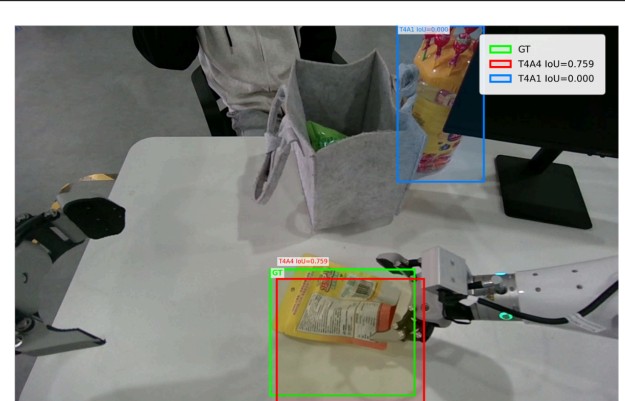

**Prompt:** Output the bounding box positions of yellow bagged food in the image.

**T4A4:** <think>\nThe yellow bagged food is located on the table, to the left of the robot arm. It is positioned near the center of the table, slightly to the right.\n</think>\n<answer>\n[546, 525, 853, 794]\n</answer>

**T4A1:** <think>\nThe yellow bagged food is located on the table, to the left of the robot arm. It appears to be a snack or candy, given its bright color and packaging. The bounding box coordinates for this object are [798, 0, 978, 323].\n</think>\n<answer>\n[798, 0, 978, 323]\n</answer>

**Discussion:** The thought in T4A4 accurately pinpoints the general location of the target object by integrating both a local perspective (to the left of the robotic arm) and a global perspective (in the middle-right of the table). In contrast, T4A1, although also using a local perspective (to the left of the robotic arm) to determine the general location of the target object, subsequently shifts focus to describing its appearance, which leads to confusion with the candy in the upper right corner.

*Figure 4.* **Case Study on Object Detection** Green text indicates key reasoning content.

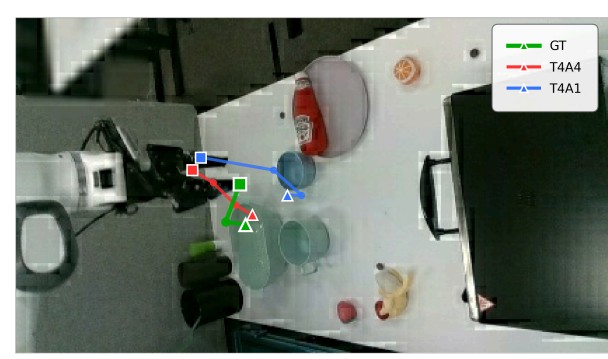

**Prompt:** Record the movement trajectory for completing reach for the strainer

**T4A4:** <think>\nTo complete the task of reaching for the strainer, the hand should first move towards the left side of the table. The strainer is located on the left side of the table, so the hand should move horizontally to the left. Once the hand is in the general area of the strainer, it should then move vertically downwards to reach the strainer. The final position should be directly above the strainer.\n</think>\n<answer>\n[[124, 120], [139, 130], [155, 148], [165, 155], [165, 155]]\n</answer>

**T4A1:** <think>\nTo complete the task of reaching for the strainer, the hand should move from its current position to the strainer. The strainer is located on the table, and the hand needs to move towards it. The hand should move horizontally to the right and slightly downward to reach the strainer.\n</think>\n<answer>\n[[130, 110], [180, 120], [200, 140], [190, 140]]\n</answer>

**Discussion:** In T4A4's thought, the target object is first localized relative to the entire tabletop; the arm's approaching direction is then derived from this global pose, and a final fine-tuning command is appended. In contrast, T4A1 merely specifies "move ... to the strainer" without first anchoring it in the scene, so the resulting terminus is biased.

*Figure 5.* **Case Study on Trajectory Prediction** Green text indicates key reasoning content.

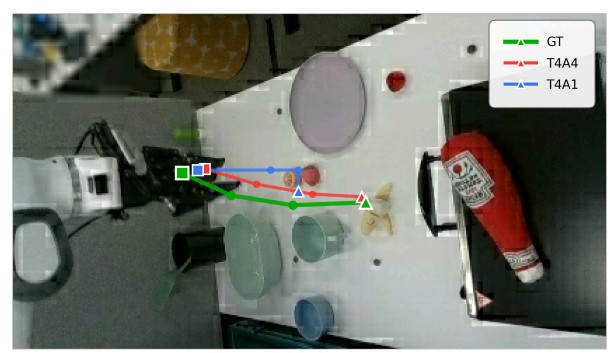

**Prompt:** What are the path coordinates of the robotic arm's end-effector when completing the "reach for the banana" task?

**T4A4:** <think>\nTo complete the task of reaching for the banana, the mechanical arm needs to move from its current position to the banana. The banana is located towards the right side of the table, slightly above the center. The arm should move horizontally to the right and slightly upwards to reach the banana.\n</think>\n<answer>\n[[135, 123], [170, 134], [210, 142], [245, 145]]\n</answer>

**T4A1:** <think>\nTo determine the path coordinates for the mechanical arm to reach the banana, we need to consider the initial position of the arm and the target position of the banana. The banana is located on the table, and the mechanical arm needs to move from its current position to the banana's position.\n\nAssuming the table is a flat surface and the banana is placed at a specific location, we can calculate the path coordinates by considering the distance and direction from the current position to the banana's position. If the table is a 2D plane, we can use the following steps to find the coordinates:\n\n1. Identify the current position of the mechanical arm.\n2. Identify the position of the banana on the table.\n3. Calculate the distance and direction from the current position to the banana's position.\n4. Use trigonometry to determine the x and y coordinates of the path.\n\nHowever, without specific pixel coordinates or a visual representation of the table, it's challenging to provide precise coordinates. If we assume the table is a standard 8x8 grid and the banana is located at a specific pixel, we can use the following approach:\n\n- If the banana is at the center of the table, the path would be a straight line from the center to the banana's position.\n- If the banana is at a specific corner, the path would involve moving diagonally or horizontally/vertically to reach the banana.\n\nFor the sake of this example, let's assume the banana is located at the center of the table, which is at (2, 2) on a 10x10 grid (0-indexed).\n\nThe path coordinates from the current position to the banana's position would be:\n[[12, 12], [18, 12], [20, 12], [20, 14]]\n\nThese coordinates represent the path the mechanical arm needs to follow to reach the banana.\n</think>\n<answer>\n[[130, 124], [180, 124], [200, 124], [200, 140]]\n</answer>

**Discussion:** In the T4A4 case, we can clearly observe that the model first identifies the location of the banana, then provides a linguistic description of the end-effector trajectory of the robotic arm, and ultimately produces a trajectory that closely resembles the ground truth. In contrast, in the T4A1 case, although the model continuously proposes schemes for determining a trajectory, it fails to accurately localize the banana. Instead, it keeps making assumptions, which ultimately leads to a significant deviation from the target in the latter part of the trajectory.

*Figure 6.* **Case Study on Trajectory Prediction** Green text indicates key reasoning content.

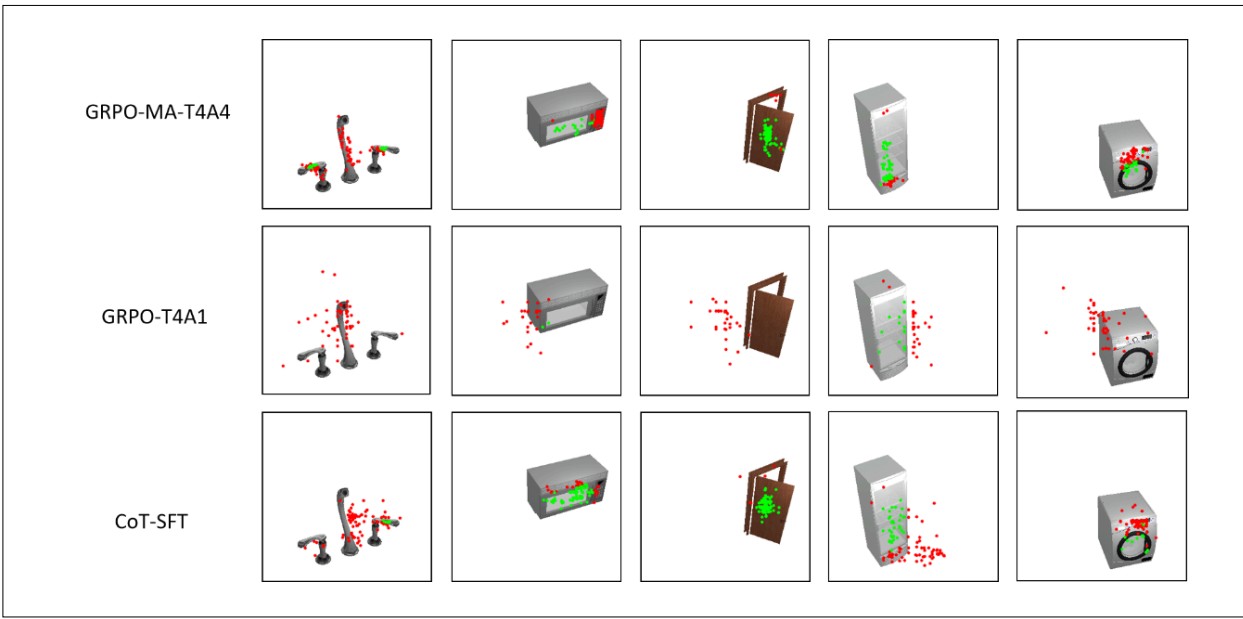

*Figure 7.* **Visualization on Simulator-based Visual Manipulation** Red dots indicate failures, while green dots represent successes. We can observe that most GRPO-MA-T4A4 points are located on the object. In contrast, GRPO-T4A1 frequently misses the object, resulting in a lower success rate.

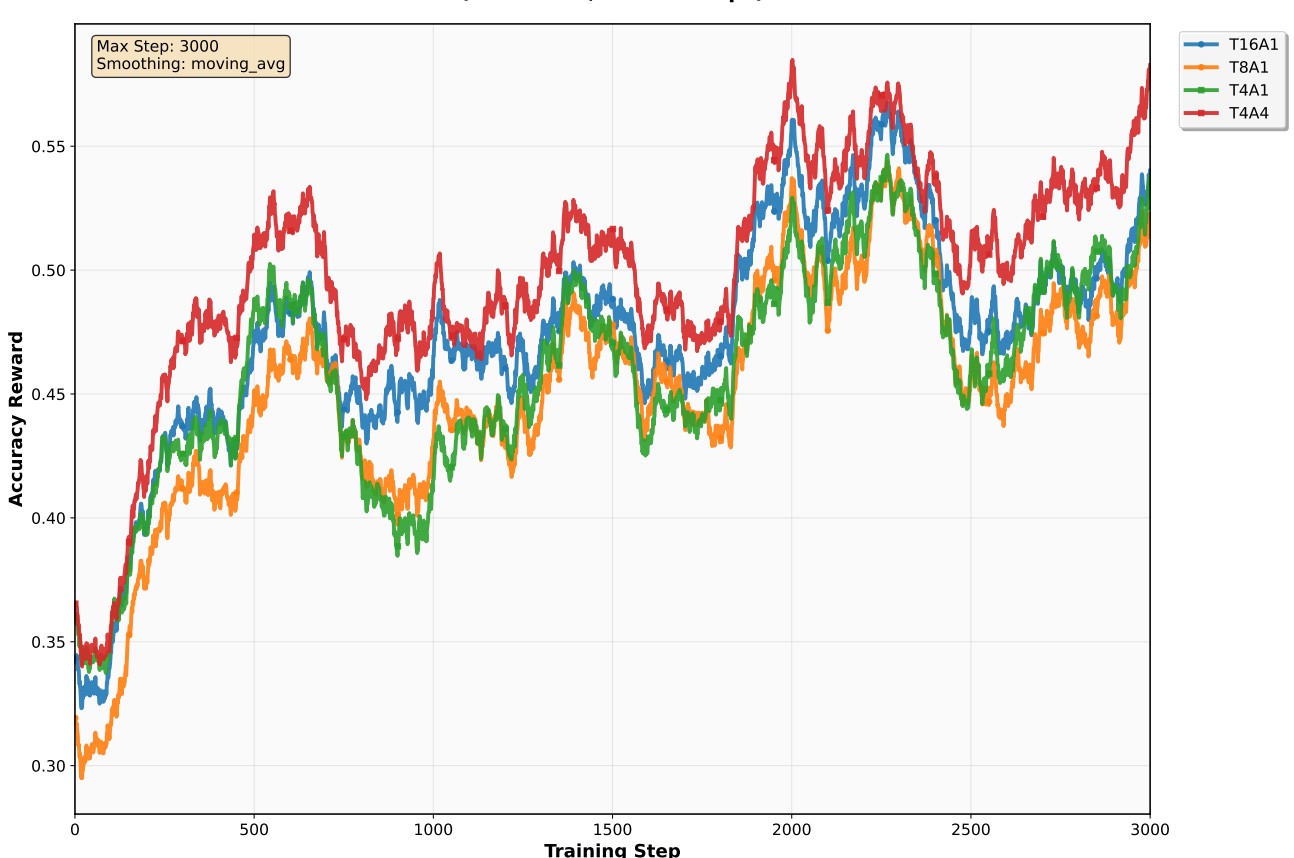

*Figure 8.* **Accuracy Reward Curve on Object Detection**

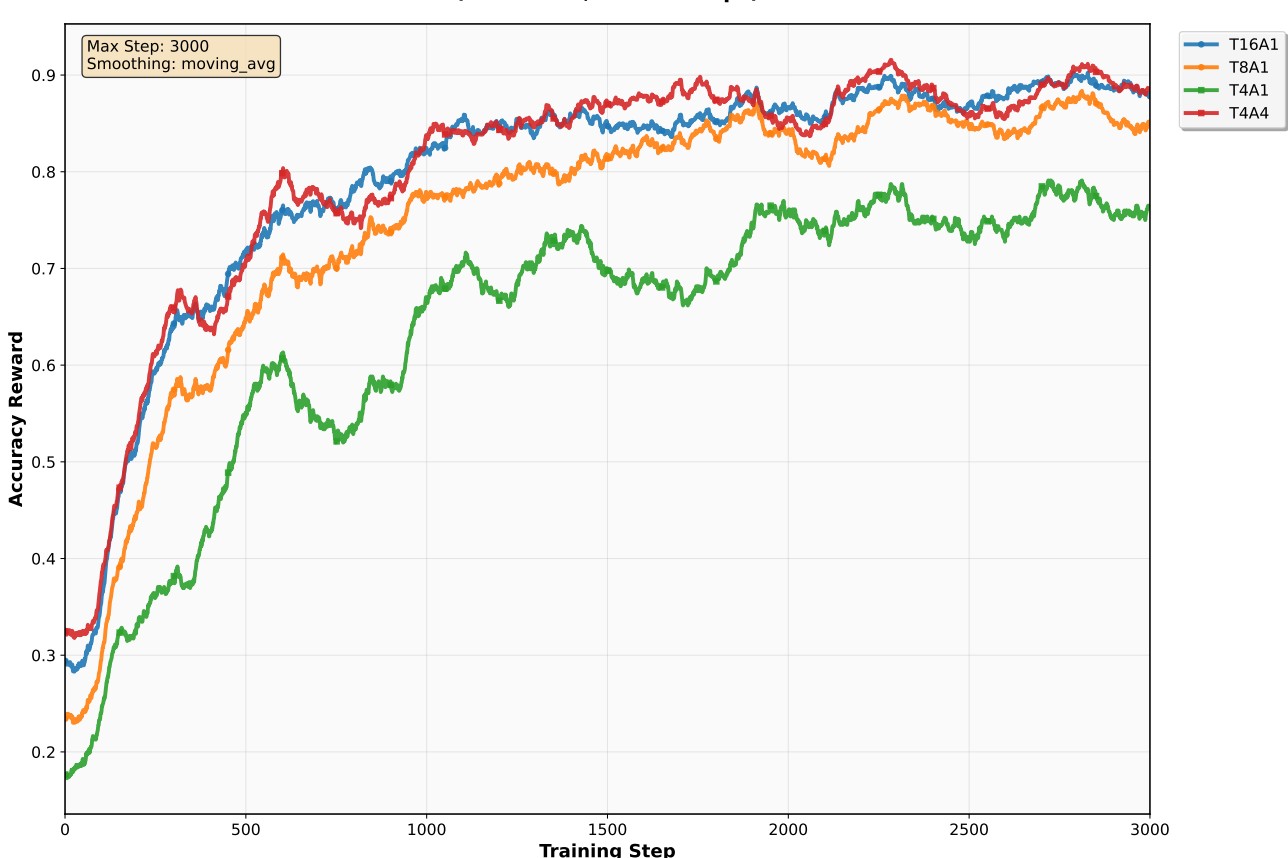

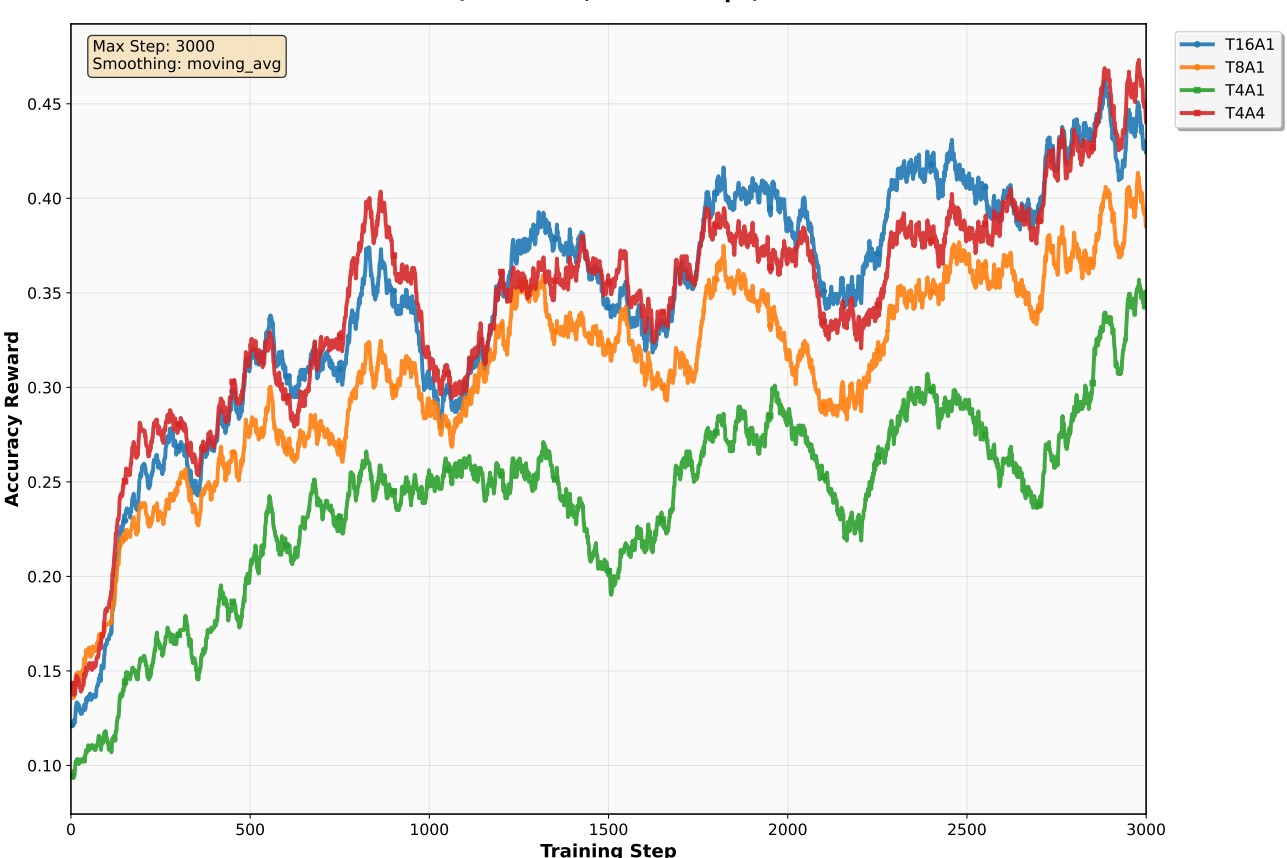

*Figure 10.* **Accuracy Reward Curve on Demand Prediction**

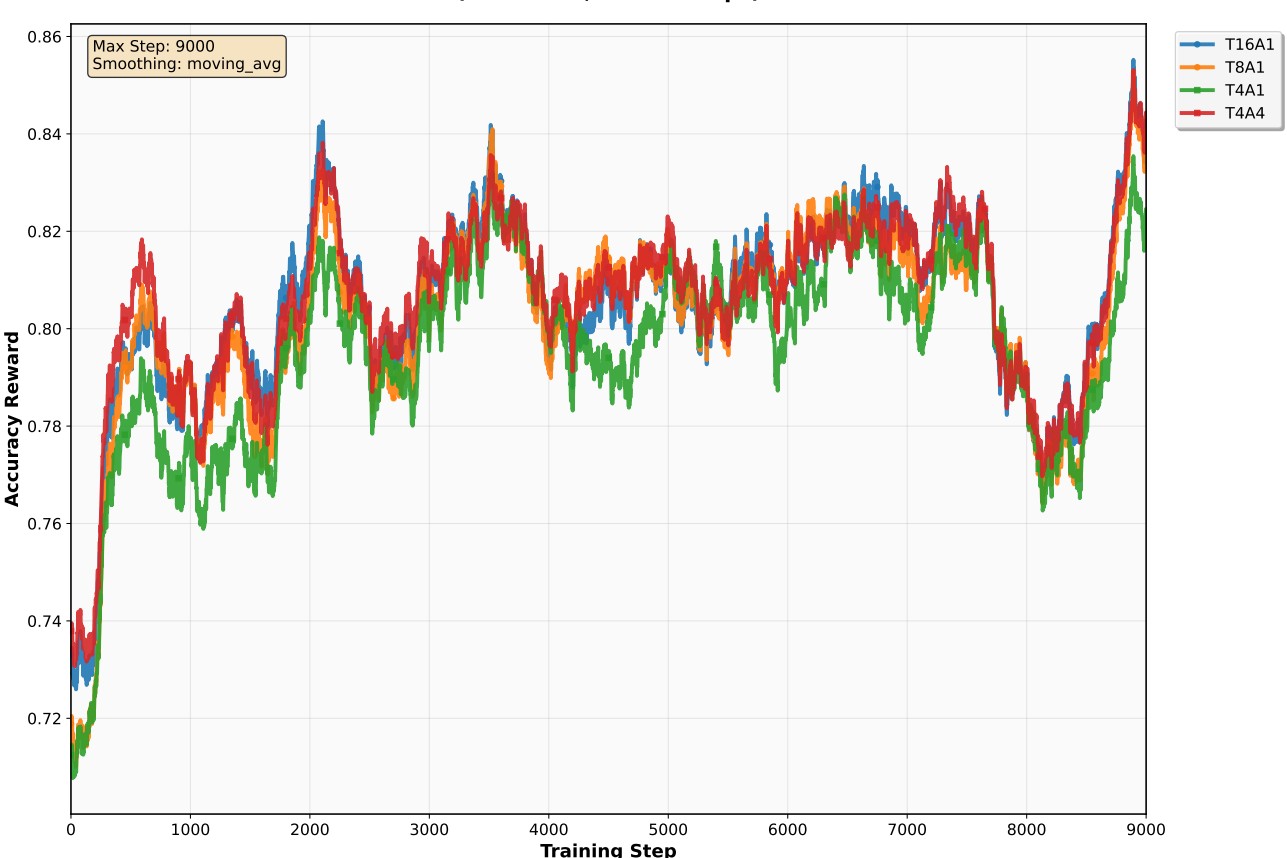

*Figure 11.* **Accuracy Reward Curve on OCR-based VQA**

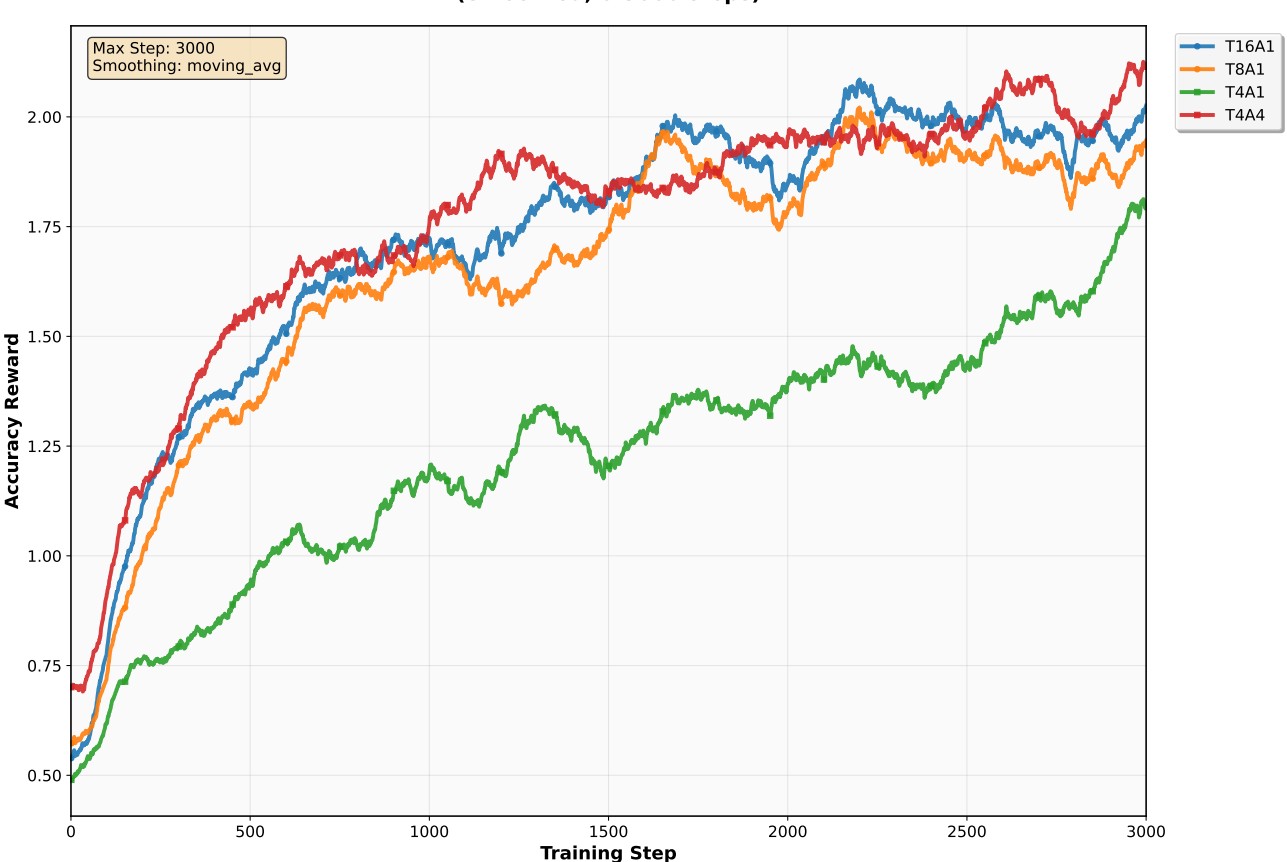

*Figure 12.* **Accuracy Reward Curve on Trajectory Prediction**

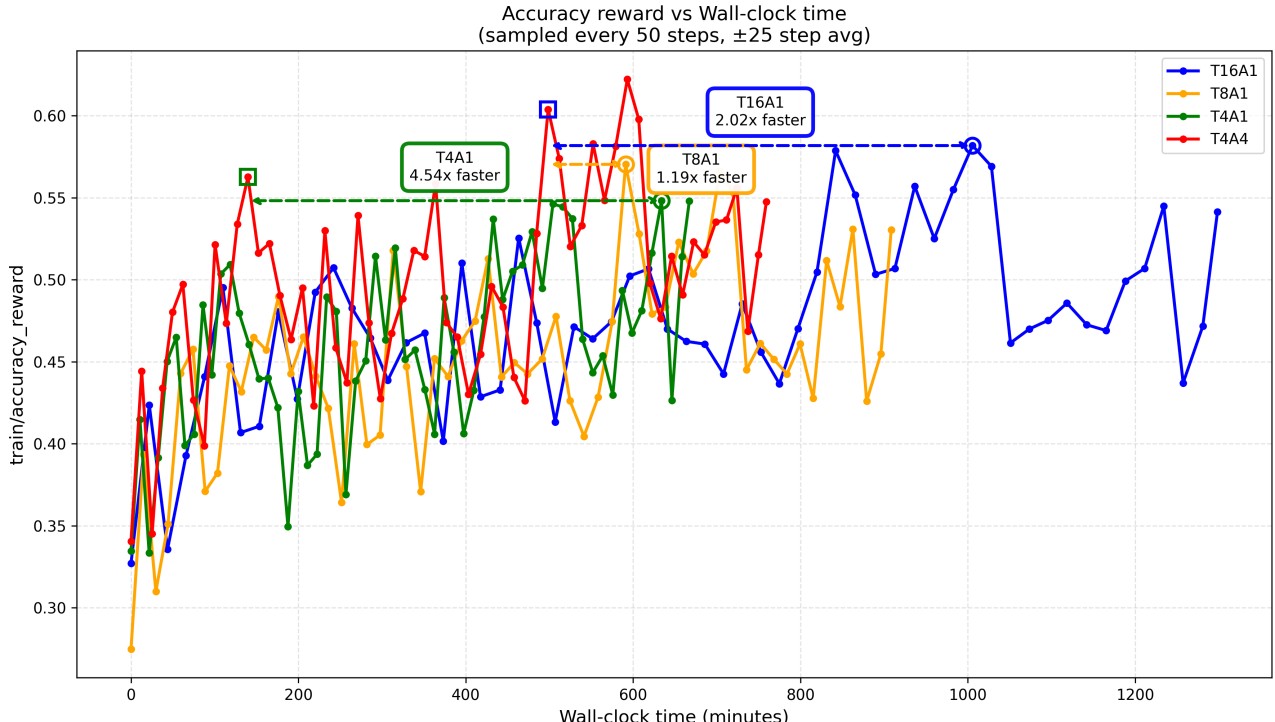

*Figure 13.* **Wall-clock Time on Object Detection**

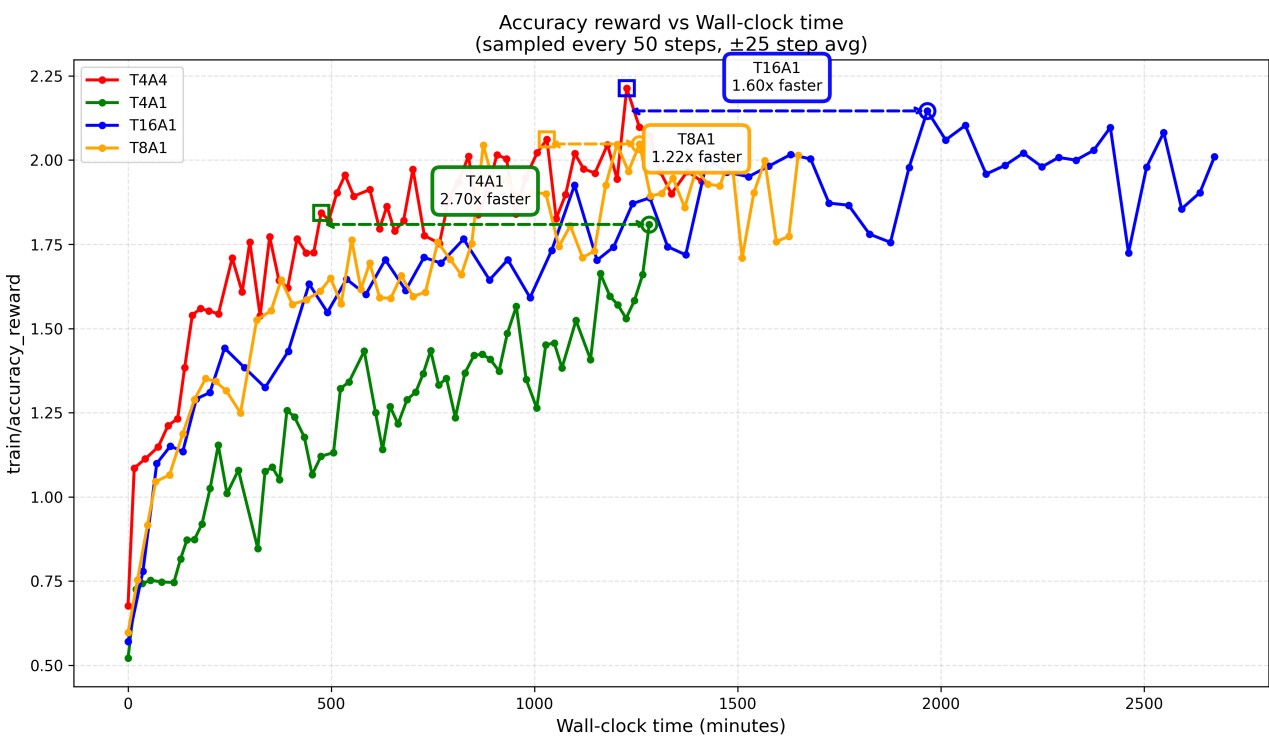

*Figure 14.* **Wall-clock Time on Trajectory Prediction**

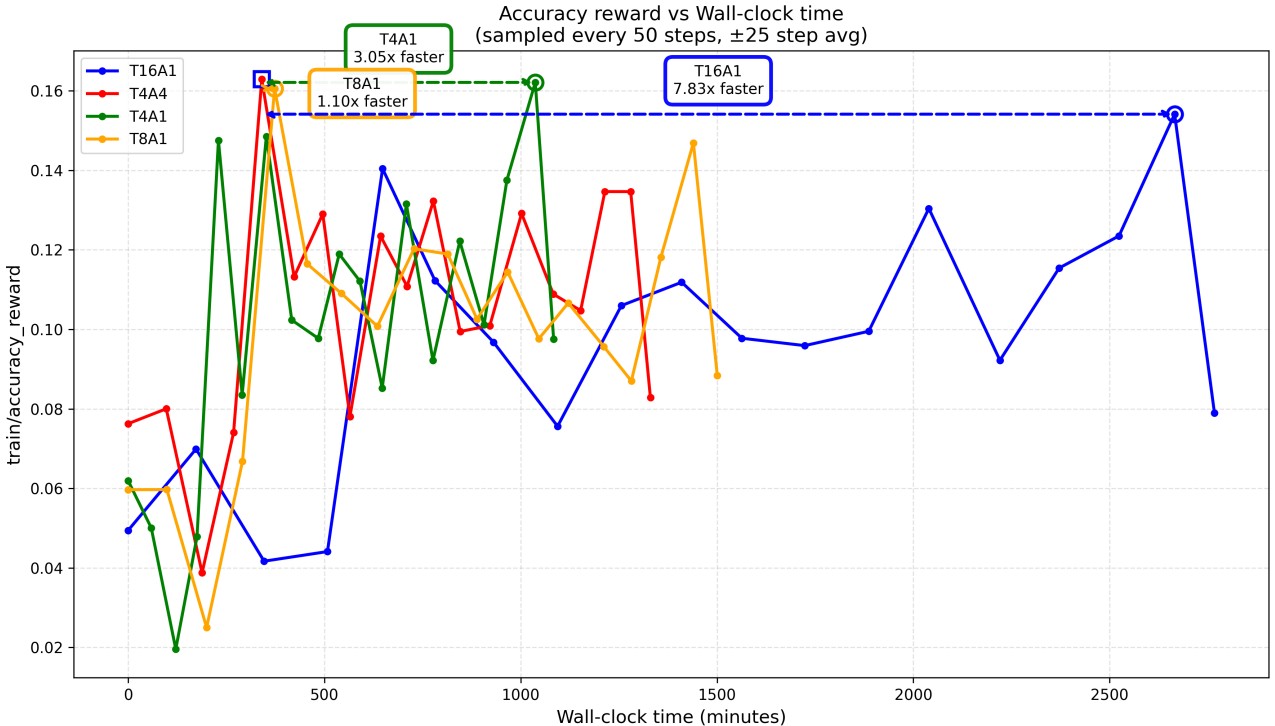

*Figure 15.* **Wall-clock Time on Code**

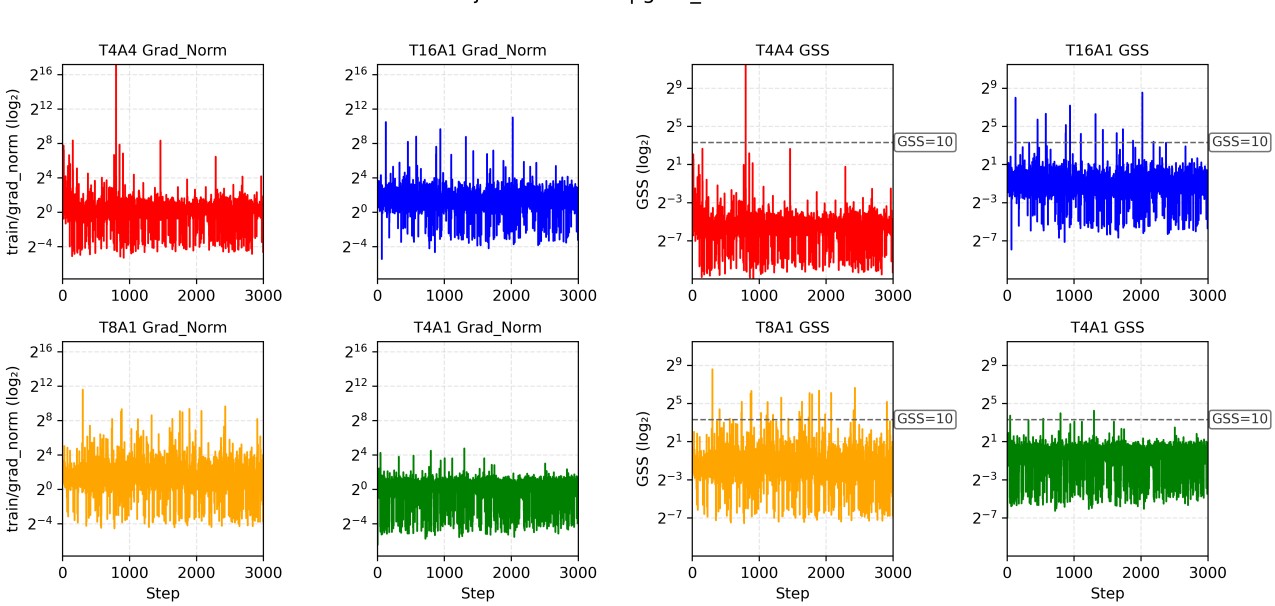

*Figure 16.* **Grad Norm and GSS on Object Detection**

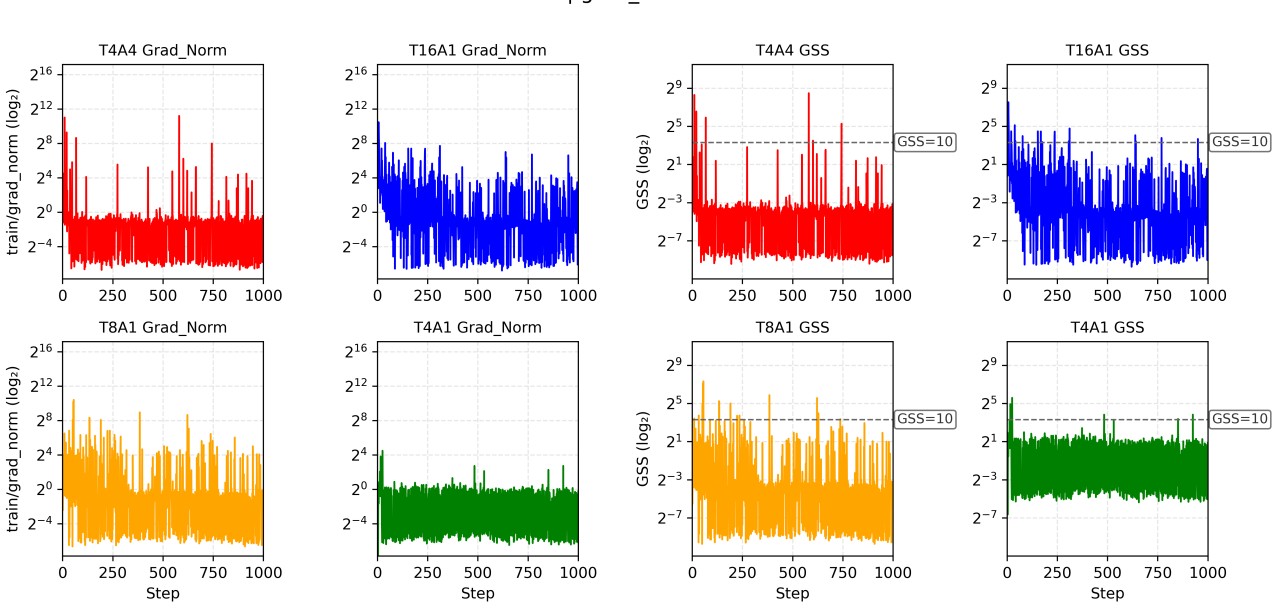

*Figure 17.* **Grad Norm and GSS on Trajectory Prediction**

*Figure 18.* **Grad Norm and GSS on Code**

*Table 14.* Additional Results on GSM8K (Math) and HumanEval (Code). TN: Number of thoughts; AN: Number of answers per thought. Bold indicates the best performance among GRPO variants.

| Model | TN | AN | GSM8K (Math) | HumanEval (Code) | |
|---|---|---|---|---|---|
| | | | Pass@1 | Pass@1 | Pass@5 |
| GRPO | 4 | 1 | 62.54 | 56.10 | 82.93 |
| GRPO | 8 | 1 | 64.67 | 56.83 | 82.93 |
| GRPO | 16 | 1 | 65.13 | 57.80 | 85.37 |
| GRPO-MA | 4 | 4 | **70.58** | **58.66** | **87.20** |

