# OpenReview forum: "Why Tree-Style Branching Matters for Thought Advantage Estimation in GRPO"
_ICML.cc/2026/Conference — ICML 2026 regular_

### Official Review · Reviewer_mBVB · 2026-03-13

**Soundness:** 1
**Presentation:** 3
**Significance:** 1
**Originality:** 3
**Overall Recommendation:** 3
**Confidence:** 3

**Summary:**

### Summary

Standard GRPO estimates thought-level advantages from a single answer per thought, which can be noisy. This work introduces GRPO-MA, which branches multiple answers from each thought and uses their average reward to evaluate thought quality separately from answer quality. A theoretical result based on the delta method argues that this branching is not just helpful but potentially necessary for stable advantage estimation. The approach is validated across eight task categories spanning text and vision domains.

**Compliance With Llm Reviewing Policy:**

Affirmed.

**Ethics Expertise Needed:**

["Legal Compliance (e.g., EU AI Act, GDPR, copyright, terms of use)"]

**Final Justification:**

I feel my concerns around the thought and answer decoupling are addressed.

**Key Questions For Authors:**

Q1. The paper sets temperature=1.0 for both thought and answer generation during training. Have the authors tried using a lower temperature (e.g., 0.1 or greedy) specifically for the answer tokens while keeping temperature=1.0 for the thought tokens? If such a setup achieves comparable performance to GRPO-MA, it would suggest that branching is mainly compensating for decoding noise rather than solving a fundamental variance problem.

Q2. The paper categorizes sigma_{R_i}^2 as "not directly controllable by the learning algorithm" (Section 4.2.3). But the sampling temperature directly controls the entropy of the answer distribution and therefore sigma_{R_i}^2. Could the authors clarify why temperature is not considered a controllable mechanism for variance reduction, given that it is a standard and easily tunable hyperparameter?  One may argue that the high temperature for answer generation is needed to maintain diversity and prevent advantage collapse, but the authors' own experiments show that thought-level expansion is more effective at this: T16A1 achieves higher NoZeroRate than T4A4 on every task in Table 10.

Q3. Table 2 reports Pass@10 and Pass@32 but not Pass@1 for math and code. Could the authors provide Pass@1 results on the same benchmarks (AIME2024, LiveBench) for all configurations including T4A1, T8A1, T16A1, and T4A4? This would clarify whether GRPO-MA improves single-sample generation quality or primarily improves coverage across many samples.

Q4. Table 8 compares GRPO-T4A1 (4 total samples) against GRPO-MA-T4A4 (16 total samples) on Qwen3-4B-Instruct. Could the authors add T16A1 to this table for a matched-budget comparison, consistent with the experimental design used in the main results (Tables 2-4)

**Limitations:**

yes

**Strengths And Weaknesses:**

### Strengths

  1. **The breadth of tasks is large.**
  Most prior work on tree-search or GRPO improvements only evaluates on math. This paper covers math, code, object detection, affordance prediction, trajectory prediction, demand prediction, OCR-based VQA, and simulator-based manipulation.

  2. **The method is simple, efficient, and practical.**
  GRPO-MA requires minimal modifications to standard GRPO and introduces virtually no new hyperparameters, making it easy to adopt in practice.

  3. **The paper is well-written and clearly organized.**
  The presentation is clear and easy to follow, and the appendix provides detailed experimental setup including prompts, reward functions, and hyperparameters, which is helpfull for reproducibility.

### Weaknesses

1. **The paper’s thought/answer decomposition is questionable.**
The whole argument seems to rely on the assumption that, even for a fixed thought, there is still substantial answer-level uncertainty. But for reasoning models, the content inside `<think>` often already contains, or at least strongly points to, the final answer. If that is the case, then thought and answer are not really cleanly decoupled, and the claimed answer-level variance may not reflect a real credit assignment issue.

2. **The “necessity” claim is confounded by stochastic decoding.**
The paper treats answer-level variance as something intrinsic, and then argues that increasing `M` is therefore necessary, i.e. "sigma_{R_i}^2 ... depend on the particular sampled thoughts and vary  stochastically across iterations." But if the thought already mostly determines the answer, then a lot of the remaining variance may just come from stochastic decoding, especially since training uses temperature 1. In that case, branching may be helping mostly because it averages out decoding noise that the training setup itself introduced, rather than because it solves some fundamentally unavoidable variance problem. So I do not think the current evidence is enough to support the stronger “necessary” claim.

3. **Table 2 omits Pass@1, which is a standard primary metric for math and code generation.**
Reporting only Pass@10 and Pass@32 makes it hard to tell whether GRPO-MA actually improves single-sample quality, or mainly improves diversity across many samples. This is also a bit inconsistent with Table 8, where Pass@1 is reported. I think Pass@1 should really be in the main math/code table.

4. **The comparison in Table 8 is not matched by total number of sampled outputs.**
GRPO-T4A1 uses 4 total sampled outputs, while GRPO-MA-T4A4 uses 16. So the improvement there cannot be cleanly attributed to the branching design alone, since part of the gain may simply come from having more sampled outputs. Given that the purpose of Table 8 is to support generalization across model architectures, I think a cleaner comparison would be something matched by total samples (or at least include T16A1, as done elsewhere in the paper).

---

> ### Author Rebuttal · Authors · 2026-03-30
>
> We sincerely thank Reviewer mBVB for recognizing the breadth of tasks, method simplicity, and clear presentation. The questions on temperature and necessity helped sharpen our arguments.
>
> ## **W1/W2: Thought/answer decomposition and the "necessity" claim.**
>
> We would first like to clarify that W1's concern — that the thought may already contain the answer — targets the branching point of **GRPO-MA** (`</think>`), which is a simple instantiation of our theory rather than the theory itself. Our core contribution is the **delta-method-based variance analysis of tree-style branching**: the proof of Eq. (3) depends only on K independent prefixes and M conditionally independent continuations, so it holds at any branching point. TreePO, TreeRL, TreeRPO all branch *within* the thought where this coupling does not arise. We apologize for not stating this corollary (generalization to arbitrary branching points) explicitly, and will add it once ICML permits — current rules do not allow manuscript updates.
>
> Nonetheless, even for GRPO-MA's specific branching point, we provide evidence that the concerns raised do not hold:
>
> **On W1 ("the content inside `<think>` often already contains the final answer").** We took steps to reduce thought-answer coupling via prompt-level soft constraints (vision: no answers in `<think>`, Appendix D; math: three-tag structure, Section 5.1). Under these constraints, per-thought reward variance reaches 38.5% of the theoretical upper bound on math, and the mean reward range (best-vs-worst answer under the same thought) spans nearly half the full scale (see our response to Reviewer ZL5C W2/Q1 for full measurements). The same thought does lead to substantially different rewards.
>
> **On W2 ("branching may be helping mostly because it averages out decoding noise").** We argue that what the reviewer calls "decoding noise" is in fact meaningful distributional information. During pretraining and SFT, the model learns p_θ = softmax(logits) at temperature=1 implicitly — this distribution encodes the model's knowledge about uncertainty and answer diversity. In the paper, the thought value V(th) = E[R(a)] where a ~ p_θ(·|th) integrates over this full distribution; multi-answer branching is Monte Carlo estimation of this quantity. Lowering temperature to T < 1 computes softmax(logits/T), collapsing the distribution: at T→0, V(th) = R(argmax p_θ(a|th)). Different temperatures define **different thought values** — lowering T reduces variance but **changes the target being estimated and loses pretraining distributional information**. The Q1 experiment below confirms.
>
> **On potentially necessary:** Scaling K alone cannot eliminate the variance floor; scaling M can — making branching the useful mechanism for driving variance toward zero in GRPO-style estimators.
>
> ## **Q1: Experiment with lower temperature.**
>
> Following Q1's suggestion, we trained GRPO-T4A1 with temp=0.1 for answer tokens (temp=1.0 for thoughts):
>
> | Setting | DFD↓ | HD↓ | RMSE↓ | EndPoint↓ |
> |---|---|---|---|---|
> | GRPO-T4A1 (temp=1.0) | 187.99 | 172.58 | 140.80 | 142.74 |
> | GRPO-T4A1 (temp=0.1) | 178.63 | 165.95 | 123.16 | 132.19 |
> | **GRPO-MA-T4A4** | **151.10** | **138.29** | **111.59** | **120.60** |
>
> Temp=0.1 improves over T4A1 by reducing answer-level variance, but underperforms GRPO-MA because it simultaneously discards distributional information (as argued in W2 above). GRPO-MA achieves both — reducing variance via branching while preserving the full distribution.
>
> ## **Q2: σ² controllability and NoZeroRate.**
>
> We agree that temperature affects σ²_{R_i}. Our phrasing "not directly controllable" was imprecise — we will revise to clarify our analysis is under a **fixed temperature**, where the only lever is sampling structure (K vs M).
>
> On NoZeroRate: avoiding advantage collapse is fundamentally a sampling efficiency question. T16A1 achieves higher NoZeroRate (i.e., fewer collapsed batches), but at ~1.6× the sampling time of T4A4. More importantly, NoZeroRate only reflects whether advantages are non-zero — it does not reflect whether advantages are *accurately estimated*. T4A4 matches or slightly outperforms T16A1 in final performance despite lower NoZeroRate, suggesting that accurate advantage estimation (via branching) matters more than simply avoiding collapse.
>
> ## **W3/Q3: Pass@1.**
>  Due to character limits, please see our response to Reviewer mJMo (W1/W2/Q1) for full tables.
>
> ## **W4/Q4: Table 8 unmatched samples.**
>
> T4A4 does *not* produce 16 independent samples — it generates 4 thoughts (same as T4A1), then branches 4 short answers from each. The improvement is **cleanly attributable to branching**: similar compute, only difference is whether branching is applied. From our main experiments, T16A1 costs ~1.6× the compute of T4A4, so it is not matched-budget. Nonetheless, we fully appreciate the reviewer's advice and commit to providing T16A1 results on Qwen3-4B shortly — GPU and character limits prevent us from including them here.

---

> > ### Author Rebuttal · Reviewer_mBVB · 2026-04-04
> >
> > I thank the authors for the thorough rebuttal. Most concerns are adequately addressed, particularly the Q1 temperature experiment and per-thought reward variance evidence. However, T16A1 costs 4× T4A1, implying T4A4 costs ~2.5× T4A1, so Table 8 remains not compute-matched.

---

> > > ### Author Response · Authors · 2026-04-04
> > >
> > > We are very grateful for the reviewer's feedback and appreciate the valuable suggestions and insightful experimental proposals. We are very pleased that additional experiments and evidence have helped alleviate some of these concerns. Since we are only allowed one round of responses after the rebuttal, we greatly value this opportunity to communicate with the reviewer.
> > >
> > >
> > > **On the compute cost comparison:** The reviewer states that T16A1 costs 4x T4A1, implying T4A4 costs ~2.5x T4A1. However, this estimate assumes training time scales linearly with the number of rollouts, which does not hold in practice. In GRPO training, the dominant cost lies in sampling rollouts, which is essentially an inference process. During inference, the prefill stage is compute-bound, while the decoding/rollout stage is memory-bound and can be efficiently parallelized [1]. As a result, increasing the number of rollouts or branches does not lead to a proportional increase in wall-clock time.
> > >
> > > Our Tables 2, 3, and 4 report Seconds per Step (S/S) for each task under different settings. The measured wall-clock time shows that T16A1 takes approximately 2x the time of T4A1, while T4A4 takes only 1.2x the time of T4A1 — far from the 2.5x the reviewer estimated. T4A4 and T4A1 therefore consume similar compute resources under the same hardware setup (4× H100 GPUs).
> > >
> > > We have also included the T16A1 experimental results below (as promised in our rebuttal) to further substantiate this point.
> > >
> > > **Math (Qwen3-4B-Instruct)**
> > >
> > > | Setting | Seconds per Step | Pass@1 |
> > > |---|---|---|
> > > | GRPO-T4A1 | 115.28 | 17.53 |
> > > | GRPO-T16A1 | 245.83 | 17.90 |
> > > | **GRPO-MA-T4A4** | 135.42 | **18.07** |
> > >
> > > Notably, GRPO-MA-T4A4 achieves comparable performance to GRPO-T16A1 (18.07 vs. 17.90) while requiring only 55% of the wall-clock time per step (135.42s vs. 245.83s). In other words, branching achieves the same level of performance at nearly half the training cost, demonstrating that it is a substantially more compute-efficient mechanism than simply scaling up the number of independent rollouts.
> > >
> > > We thank the reviewer for this suggestion and agree that including T16A1 strengthens the experimental comparison. We will add T16A1 to Table 8 along with S/S measurements in the revised manuscript.
> > >
> > > [1] DistServe: Disaggregating Prefill and Decoding for Goodput-optimized Large Language Model Serving

---

### Official Review · Reviewer_VqWs · 2026-03-13

**Soundness:** 4
**Presentation:** 2
**Significance:** 3
**Originality:** 4
**Overall Recommendation:** 5
**Confidence:** 3

**Summary:**

The submission uses the multivariate delta method to analyze the asymptotic variance of the GRPO group advantage estimator as a function of the number of CoT traces sampled and (separately) the number of answers sampled conditioned on a CoT trace. It uses it to establish that merely sampling more CoT traces is insufficient relative to sampling more answers per CoT trace, and then shows the empirical benefit of the approach through a variety of experiments.

**Compliance With Llm Reviewing Policy:**

Affirmed.

**Final Justification:**

I still think this is a solid paper. Most of my comments have to do with clarity and I think that work could easily be done in rebuttal as the authors promise. I think a stronger paper would show stronger empirics and/or perhaps more sophisticated branching strategies, hence not rating any higher.

**Key Questions For Authors:**

N/A but see the review weaknesses for things that could be improved.

**Limitations:**

No concerns.

**Strengths And Weaknesses:**

# Strengths

The submission adds to the body of work reducing the variance of the GRPO group advantage estimator, which is of obvious use given the prevalence of GRPO-like methods in current practice, and the analysis is new to my knowledge. The empirical results are extensive and consistently favorable to GRPO-MA (though modest in magnitude for some tasks, e.g. affordance dataset, VQA). I also have not seen delta method applied this in this domain before (I think it is more common closer to stats, and I've mostly thought about advantage estimation from a control variates perspective before reviewing this paper), so I am hopeful that it motivates further advances / extensions.

# Weaknesses

While the writing is solid, the structure is such that I took a few readthroughs of the paper to understand what is going on. Some examples of my confusion:


1. "Thought advantage estimation" isn't something I've seen used as a term of art -- I think here it is still used operationally as "advantage estimation for everything before the </think> tag" but formally I can't quite tell if it is meant to be "any set of tokens that are used to condition more than one graded reward" or something like this. Basically is there a definition for it that's more than just the "we sample k traces and then m answers conditional on each trace, and the first set of things is the thoughts"?
2. Similarly I think a reader should be given a little more introduction into the delta method. I think it might be more common in the stats literature than deep learning, so even a basic formal statement explaining what it is, or (better) an informal intuition / explanation in terms of the CLT / local linearization / variance propagation etc might help the ICML audience before showing the variance result. Furthermore, being explicit on what the `\approx` means (i.e. it is asymptotic in M and converges in distribution) in the main text would be useful.
3. Related to the above, a lot of significant details are shunted to the supplement, which is 3 times longer than the main text and includes critical information like the core derivation, description of all the experimental evals, training details, etc. Meanwhile, the main text spends a substantial amount of its space on the introduction and related work, as well as results tables. I think a better organization would tighten the introduction, surface a clean and self-contained statement of the core results (the variance estimate and sketch of how it is derived), and prioritize the strongest empirical results (including enough details to understand them). Then it could shift to the appendix additional discussion of related work, as well as supplementary experiments where GRPO-MA's results are more modest. Also on this front -- especially when effect sizes are tiny (e.g. 42.63 for GRPO-MA vs 42.47 for GRPO on demand prediction accuracy, or 94.22 vs 94.20 and 76.69 vs 76.65 for VQA), the paper should report error bars and convince us these tiny effects are real and not noise.
4. I found missing additional intuition on "what's doing the work" in the variance result. Something about having multiple reward draws for the same prefix is important, but the paper doesn't help me understand why. One intuition would come from a control variates perspective, where the group baseline needs to be as correlated as possible with the current value while being independent of the action -- in that case it should be obvious that sharing more of the prefix can increase the correlation. And perhaps decomposing expression 3 more explicitly into the value and baseline would help show this and provide additional insight. Maybe there are other ways of viewing the result to improve intuition and understanding.

---

> ### Author Rebuttal · Authors · 2026-03-30
>
> We sincerely thank Reviewer VqWs for the constructive suggestions, especially the control variates perspective, the call for clearer delta method intuition, and the advice on paper organization — we are grateful for these insights.
>
> ## **W1: "Thought advantage estimation" definition.**
>
> We fully agree and apologize for the imprecise terminology. We provide a formal definition below.
>
> **Definition (Shared-Context Advantage Estimation).** Given a generation sequence split into a shared context $s$ and a continuation $a$, *shared-context advantage estimation* refers to estimating the advantage of $s$ by sampling multiple continuations $\{a_1, \dots, a_M\}$ conditioned on $s$, using their rewards or values to estimate the context value $V(s) = \mathbb{E}_{a \sim \pi(\cdot|s)}[R(s, a)]$, and normalizing within the group to obtain $A(s)$. "Thought advantage estimation" as used in our paper is the special case where $s$ corresponds to the tokens inside `<think>...</think>` (vision/code) or `<analysis>...</analysis>` (math).
>
> Importantly, this formal definition is not tied to the `</think>` delimiter — any partition of the generation into a shared context and multiple continuations falls under it. TreePO, TreeRL, and TreeRPO all perform shared-context advantage estimation at different granularities. Furthermore, Eq. (3) generalizes directly: replacing "thought" with "shared context" yields an identical variance expression under the same three conditions (independent contexts, conditionally independent continuations, per-continuation evaluation).
>
> **Revision plan:** We will (1) add this formal definition in Section 4; (2) add an explicit corollary with the generalized variance formula for arbitrary branching points; (3) add a remark connecting this to existing tree-style methods, clarifying that our theory provides a unified explanation for why they achieve variance reduction.
>
> ## **W2/W4: Delta method introduction and intuition for the variance result.**
>
> We thank the reviewer for these suggestions, which we find particularly insightful. The control variates perspective (W4) offers an intuition that we had not explicitly articulated in the manuscript, and we believe it will help readers understand the result more naturally. Both points concern the lack of accessible intuition for why branching reduces variance, and addressing them will meaningfully improve readability. In the revised manuscript, we will add an intuition-focused paragraph in Section 4 that: (1) introduces the delta method informally (CLT → local linearization → variance propagation, and clarifying that "≈" arises from the first-order Taylor linearization in the delta method); (2) connects the formal result to the control variates perspective the reviewer raised in W4 — explaining *why* sharing more of the prefix helps, not just *that* it helps. We believe combining the formal delta method analysis with the intuitive control variates view will make the result substantially more accessible to the ICML audience.
>
> ## **W3: Paper organization and error bars.**
>
> We fully agree that the current balance between main text and supplement is suboptimal. The revised manuscript will: (1) tighten the introduction and related work to reclaim space; (2) surface a self-contained derivation sketch of the core variance result (Eq. (3)) into the main text, so readers can follow the key argument without referring to the appendix; (3) Adjust the layout of the result report.
>
> On error bars: we appreciate this concern, especially for small effect sizes (e.g., demand prediction 42.63 vs 42.47, VQA 94.22 vs 94.20). We would like to clarify that the primary controlled comparison is T4A4 vs T4A1 (similar compute, with/without branching), where gains are larger and consistent across all tasks. The T4A4 vs T16A1 comparison serves a different purpose: demonstrating that at ~60% of T16A1's per-step cost, T4A4 achieves comparable performance — so the modest margins here reflect a favorable efficiency trade-off. We will add error bars or other equivalent methods for results with small effect sizes and revise "slightly outperforms" to "matches or slightly outperforms" for T4A4 vs T16A1 to more accurately reflect the results.

---

> > ### Author Rebuttal · Reviewer_VqWs · 2026-03-31
> >
> > I appreciate the authors' response and promise to reorganize the paper, and think the shared-context advantage definition is much clearer. It also perhaps opens up other notions of sharing context for variance reduction beyond the one considered here.
> >
> > In there being a fairly large restructure promised in the camera-ready and already rating the paper at a 5, I don't feel comfortable raising my score further, but nonetheless remain positive on the paper.

---

> > > ### Author Response · Authors · 2026-04-01
> > >
> > > We sincerely thank the reviewer for the continued support and the valuable suggestions throughout this discussion. The reviewer's suggestions are so valuable that they will greatly improve the readability of our paper.

---

### Official Review · Reviewer_ZL5C · 2026-03-14

**Soundness:** 3
**Presentation:** 2
**Significance:** 2
**Originality:** 2
**Overall Recommendation:** 4
**Confidence:** 3

**Summary:**

The paper proposes to sample multiple answers per thought in GRPO training. They show that increasing just the number of thoughts leads to a variance estimator that depends on the variance of the thought, while increasing the number of sampled answers per though reduces the variance of the estimator. They show in experiments (primarily using Qwen vision-language models) that their approach outperforms GRPO and SFT baselines.

**Compliance With Llm Reviewing Policy:**

Affirmed.

**Final Justification:**

The rebuttal addressed my concerns and I have increased my score from a 3 to a 4.

**Key Questions For Authors:**

1. In practice, across some of the tasks investigated, how much variance is there in the answer given a thought?
2. Why is the performance gains on language modelling tasks minimal on math and coding tasks, whereas the gains on the vision-language model were larger?

**Limitations:**

Yes

**Strengths And Weaknesses:**

Strengths:
- The authors performed a large number of empirical experiments, in particular on vision language models, to validate their proposed approach of increasing the number of answers per thought. Their approach outperformed GRPO and SFT baselines.
- The authors showed that increasing the number of answers per thought reduces the variance for the advantage

Limitations:
- for certain tasks the benefit of the proposed approach is minimal, for example on the pure language modeling tasks (Sect 5.5) (even when sampling additional answers per question).
- It would be helpful if the authors could better articulate why they expect variance in the answer given the thought -- for example if the variance in practice in negligible then sample a single answer given the though would be sufficient. Some empirical analyses of these quantities could strengthen the paper.

---

> ### Author Rebuttal · Authors · 2026-03-30
>
> We sincerely thank Reviewer ZL5C for acknowledging the extensive experiments and the variance reduction results. The question on answer-level variance (W2/Q1) prompted us to conduct additional analyses that we believe strengthen the paper.
>
> ## **W1/Q2: Minimal benefit on pure language tasks.**
>
> We would like to first draw the reviewer's attention to the GSM8K and HumanEval results already reported in our appendix (Table 11), where the gains on language tasks are more pronounced:
>
> | Benchmark | T4A1 | T16A1 | **T4A4** |
> |---|---|---|---|
> | GSM8K Pass@1 | 62.54 | 65.13 | **70.58** |
> | HumanEval Pass@1 | 56.10 | 57.80 | **58.66** |
>
> One possible explanation for the seemingly smaller gains on math/code in the main paper is benchmark difficulty. The main paper reports on AIME 2024 and LiveBench — highly challenging benchmarks where absolute scores are low, making meaningful differences harder to observe. In contrast, on GSM8K T4A4 outperforms T4A1 by **+8.04** and T16A1 by **+5.45**; on HumanEval T4A4 outperforms T4A1 by **+2.56** and T16A1 by **+0.86** — gains that are clearly visible and consistent. This suggests the method works across modalities, but benchmark difficulty compressed the visible gains in the main paper.
>
> Across all evaluated domains — manipulation, vision, math, and code — T4A4 consistently outperforms flat-sampling baselines (T4A1, T8A1) and achieves comparable or slightly better performance than T16A1 at a fraction of its compute cost (T4A4 costs only 1.1–1.3× the per-step time of T4A1, while T16A1 costs 1.8–2.4×, see Section 5.4). This universality is what our variance reduction theory predicts. We also note that GRPO-MA is a deliberately *minimal* branching instance designed to validate our theory, not a performance-optimized algorithm. Our theory applies equally to other tree-style methods (e.g., TreeRL, TreePO), since The proof of Eq. (3) only requires K independent prefixes and M conditionally independent continuations — a structure shared by all tree-style branching regardless of where branching occurs. We will add this general corollary in the revised manuscript. More sophisticated branching strategies may yield larger gains and are an important future direction.
>
> ## **W2/Q1: Why expect variance in the answer given the thought, and how large is it?**
>
> **1. Direct measurement of answer reward variance.** To directly quantify how much variance exists in the answer given the thought, we conduct the following measurement: for each training sample, we fix the thought prefix and sample M=4 independent answers using the trained model at temperature=1.0. We then compute the reward for each answer using the task-specific reward function, and report two statistics averaged across all samples: (1) **Mean Variance** — the average of per-sample reward variances $\text{Var}(R(a_1), \dots, R(a_M))$; (2) **Mean Range** — the average of per-sample reward ranges $\max_j R(a_j) - \min_j R(a_j)$, which provides an intuitive measure of how much the best and worst answers differ under the same thought. All rewards are normalized to [0, 1].
>
> | Metric | Math | Object Detection | Trajectory Prediction |
> |---|---|---|---|
> | Reward Type | Binary (0/1) | Continuous (IoU) | Continuous |
> | Mean Variance | 0.0963 | 0.0577 | 0.0463 |
> | Mean Range (max−min) | 0.4660 | 0.2972 | 0.2158 |
>
> For rewards in [0,1], the theoretical upper bound of variance is 0.25 (achieved only by the extreme case of Bernoulli(0.5)). Math's mean variance of 0.0963 already reaches **38.5%** of this upper bound — a substantial fraction. The mean range further illustrates this: even for the most stable task (Trajectory), the best-vs-worst answer under the *same thought* spans over one-fifth of the full reward scale; for Math, this gap approaches half. These are not negligible fluctuations — they directly ground the $\sigma^2_{R_i}$ term in our theory.
>
> **2. Training statistics confirm non-negligible variance.** If answers given a thought were near-deterministic, we would expect two things: (a) the **inconsistency rate** (defined in Appendix G.1) should be close to zero, since all answers would receive similar rewards and thus agree on the gradient direction; (b) the **NoZeroRate** (defined in Appendix G.3) should be similar between T4A4 and T4A1, since multi-sampling would not produce diverse outcomes. Neither holds in practice. The inconsistency rate is **25.65%** (trajectory) and **24.83%** (object detection) under T4A4 (Appendix G.1) — roughly one in four answers has a gradient direction conflicting with its thought's advantage. T4A4 also achieves substantially higher NoZeroRate than T4A1 across all binary-reward tasks (Table 10). Both observations confirm that answer-level variance is genuine and non-negligible.
>
> We will include the variance analysis as a new table in the revised paper.

---

> > ### Author Rebuttal · Reviewer_ZL5C · 2026-04-04
> >
> > I thank the authors for their detailed reply. I appreciate the analysis of the variance in the answer given the thought, and also the related response to reviewer mBVB, which analyzed the related question of reducing the temperature of the answer generation as opposed to increasing the the number of generations. The authors also make a good point of the potential applicability of the approach beyond GRPO. The authors have adequately addressed my concerns and I will increase my score from 3 to 4.

---

> > > ### Author Response · Authors · 2026-04-04
> > >
> > > We sincerely thank the reviewer for the time and effort devoted to this follow-up and for recognizing the value of our additional analyses on answer-level variance, the comparison with lowering answer-generation temperature, and the broader applicability beyond GRPO. We are grateful that our response has addressed the reviewer's concerns.

---

### Official Review · Reviewer_mJMo · 2026-03-18

**Soundness:** 3
**Presentation:** 3
**Significance:** 2
**Originality:** 2
**Overall Recommendation:** 4
**Confidence:** 4

**Summary:**

This paper proposes GRPO-MA, a tree based adaption of the standard GRPO algorithm to sample multiple answers (solutions) for each of the intermediate thinking traces. This results in two stage branching where the first level has K nodes (corresponding to K thoughts as in standard GRPO) and an additional second level unlike GRPO to sample M solutions each for each of the thinking traces resulting in $K \times M$ total solutions. This helps in estimating the "value" of each of the intermediate thoughts by using GRPO-like advantage estimation and further breaking down the GRPO objective into thought (normalized by K thoughts) and solution tokens (normalized by $KM$ total answers).

The authors further provide a theoretical analysis showing that scaling solutions per thought reduces the variance by $O(1 / M)$ where M is the number of sampled solutions per thought. The authors show a lot of empirical results on Qwen-3-VL-3B-Instruct model across a bunch of tasks comparing with the GRPO baseline.

**Compliance With Llm Reviewing Policy:**

Affirmed.

**Final Justification:**

I am increasing my score from 3 to 4 after the authors' rebuttal. It's still not a very strong contribution for ICML which is why I am giving it a weak accept.

**Key Questions For Authors:**

- I would like to see `pass@1` scores for math/code for Tables 2 and 3 as well.
- I would also like to see the comparison with other tree-based methods like TreeRL and TreePO.
- GRPO is shown to be a bit unstable in latest works like DAPO [1], CISPO [2], and ScaleRL [3]. Does the proposed tree-based approach work on these stronger algorithms?

[1] DAPO: An Open-Source LLM Reinforcement Learning System at Scale, Yu et. al, 2025

[2] MiniMax-M1: Scaling Test-Time Compute Efficiently with Lightning Attention, Chen et. al. 2025

[3] The Art of Scaling Reinforcement Learning Compute for LLMs, Khatri et. al, 2025.

**Limitations:**

Yes

**Strengths And Weaknesses:**

Strengths
- The work is clearly motivated, the modifications to GRPO are simple and intuitive, and the theory is strong.
- The empirical results cover a lot of tasks and clear detailed training setup and hyperparameters are provided for each of the tasks are provided in the appendix.
- The GSS analysis shows the stability of GRPO-MA compared to GRPO.

Weaknesses
- I believe the empirical results are not super impressive for GRPO-MA (T=4,A=4) compared to GRPO (T=16,A=1) or for code even GRPO (T=4,A=1). I know GRPO (T=16,A=1) is computationally more expensive given thought length is usually higher.
- Also Tables 2/3 do not report `pass@1` for math/code for the RL methods. Is there a reason behind that?
- Training curves for math/code are missing from the analysis, so I can't judge how noisy the methods are.
- Stronger baselines like TreePO are missing

---

> ### Author Rebuttal · Authors · 2026-03-30
>
> We sincerely thank Reviewer mJMo for recognizing the clear motivation, strong theory, and comprehensive GSS analysis.
>
> ## **W1/W2/Q1: Empirical results and missing Pass@1.**
>
> We would like to clarify that the core contribution of this paper is to use variance analysis to theoretically demonstrate the importance of tree-style branching for GRPO, rather than to propose a state-of-the-art branching algorithm. GRPO-MA is deliberately designed as a *minimal* tree-style branching instance to validate our theory.
>
> We now report Pass@1/2/5 for AIME/LiveBench (n=100) and Pass@1 for GSM8K/HumanEval (n=10).
>
> | Benchmark | T4A1 | T8A1 | T16A1 | **T4A4** |
> |---|---|---|---|---|
> | AIME Pass@1 / @2 / @5 | 2.07 / 3.79 / 7.64 | 2.20 / 3.92 / 7.39 | 2.93 / 5.19 / 9.66 | **3.40 / 5.93 / 10.74** |
> | GSM8K Pass@1 | 62.54 | 64.67 | 65.13 | **70.58** |
> | LiveBench Pass@1 / @2 / @5 | 6.59 / 8.46 / 10.46 | 6.53 / 8.42 / 10.24 | 6.30 / 8.10 / 10.06 | **6.73 / 8.73 / 10.57** |
> | HumanEval Pass@1 | 56.10 | 56.83 | 57.80 | **58.66** |
>
> T4A4 outperforms T4A1 (similar compute, without branching) by substantial margins — e.g., **+8.04** on GSM8K, **+1.33** on AIME Pass@1 — and matches or slightly outperforms T16A1 at only 1.1–1.3× the cost of T4A1, versus 1.8–2.4× for T16A1. For code, improvement is more modest but consistent across Pass@1/2/5. We highlight that the key empirical observation is not the absolute magnitude of any single gain, but rather the **consistency of improvement across all eight task categories** — this universality is precisely the signature predicted by our variance reduction theory: more accurate advantage estimation yields healthier gradient signals regardless of task, and no task-specific tuning is needed.
>
> ## **W3: Training curves.**
>
> Unfortunately, ICML does not allow revised manuscripts during rebuttal. We will add training curves once revisions are permitted, and further open-source our datasets.
>
> Regarding noise, the paper provides another evidence of training stability: the GSS analysis (Appendix G.2) confirms GRPO-MA maintains more stable gradient signals than baselines — a measurement of training-time noise. The grad norm curves for code (Figure 18) further corroborate smoother optimization. The consistent improvements across benchmarks of varying difficulty (GSM8K, HumanEval, AIME, LiveBench above) further suggest robustness.
>
> ## **W4/Q2: TreePO/TreeRL.**
>
> We clarify that our goal is not to propose the optimal branching strategy, but to theoretically analyze *why* tree-style branching helps. GRPO-MA is the simplest instantiation to facilitate this analysis — not to compete with more sophisticated methods. With this in mind, we have included TreeRPO (another tree-style method) in Section 5.3 alongside non-tree improvements, showing **tree-style methods outperform non-tree ones**. To further alleviate this concern, we ran TreePO on trajectory prediction:
>
> | Method | DFD↓ | HD↓ | RMSE↓ | EndPoint↓ |
> |---|---|---|---|---|
> | GRPO-T4A1 (non-tree) | 187.99 | 172.58 | 140.80 | 142.74 |
> | TreePO | **150.28**| **136.20** | 113.11| **119.43** |
> | GRPO-MA-T4A4 | 151.10 | 138.29 | **111.59** | 120.60 |
>
> Two observations. First, **both tree-style methods substantially outperform the non-tree baseline** (~20%), which is the central claim of our theory. Second, TreePO achieves comparable or slightly better performance than GRPO-MA despite using a completely different branching strategy (sub-tree grouping vs branching at `</think>`). **Importantly, TreePO still falls within our variance analysis framework**: The proof of Eq. (3) does not depend on the branching location — it only requires K independent prefixes and M conditionally independent continuations, a structure shared by all tree-style methods (we will add this general corollary in the revision). TreePO's slight edge likely comes from its additional sampling budget reallocation mechanism, which is orthogonal to the variance reduction our theory characterizes.
>
> We apologize for not running TreeRL due to time and GPU constraints.
>
> ## **Q3: DAPO, CISPO, ScaleRL.**
>
> Regarding DAPO: we have already incorporated its key techniques (clip higher, token-level PG loss) into our GRPO implementation used in **all** main experiments — the first two rows below are the same results reported in the main paper (Table 4). CISPO's core technique (importance sampling clipping) is also used in MiniMax-M1 and ScaleRL. We additionally ran CISPO + branching (last two rows). Results on trajectory prediction:
>
> | Base Algorithm | Setting | DFD↓ | HD↓ | RMSE↓ | EndPoint↓ |
> |---|---|---|---|---|---|
> | GRPO + DAPO tricks | T4A1 | 187.99 | 172.58 | 140.80 | 142.74 |
> | GRPO + DAPO tricks | **T4A4** | **151.10** | **138.29** | **111.59** | **120.60** |
> | GRPO + DAPO + CISPO | T4A1 | 175.29 | 161.10 | 129.27 | 133.26 |
> | GRPO + DAPO + CISPO | **T4A4** | **142.25** | **129.78** | **106.77** | **108.99** |
>
> Tree-style branching remains effective on top of both.

---

> > ### Author Rebuttal · Reviewer_mJMo · 2026-04-03
> >
> > Thank you for the authors response, I will increase my score.

---

> > > ### Author Response · Authors · 2026-04-03
> > >
> > > Thanks for your thoughtful feedback and for taking the time to carefully consider our rebuttal. We are very glad that our response alleviated your concerns. We sincerely appreciate your recognition of our work and its contributions, particularly in terms of motivation, and empirical analysis.

---

### Decision · Program_Chairs · 2026-04-30

**Decision:**

Accept (regular)

**Comment:**

This paper was reviewed by four experts in the field.  The reviewers indicate that the paper is clearly motivated, with simple and intuitive modifications to GRPO, and extensive empirical results across many tasks. The method is considered simple, efficient, and practical, and effectively reduces the variance of the advantage estimator, improving stability. However, the reviewers also note some weaknesses, including not super impressive empirical gains, missing pass@1 results, lack of stronger baselines, and questions about the assumption of answer-level variance.
Based on the feedback from reviewers, the decision was made to recommend it for acceptance. We congratulate the authors on their acceptance!